# Vitamin C epigenetically controls osteogenesis and bone mineralization

Roman Thaler [1,2,3] ✉, Farzaneh Khani[1], Ines Sturmlechner [4], Sharareh S. Dehghani[1], Janet M. Denbeigh[1], Xianhu Zhou[1], Oksana Pichurin[1], Amel Dudakovic[1,2], Sofia S. Jerez[1,2], Jian Zhong [5], Jeong-Heon Lee [2,5], Ramesh Natarajan[6], Ivo Kalajzic[7], Yong-hui Jiang [8], David R. Deyle[9], Eleftherios P. Paschalis[10], Barbara M. Misof [10], Tamas Ordog [5,11] & Andre J. van Wijnen[12] ✉

Vitamin C deficiency disrupts the integrity of connective tissues including bone. For decades this function has been primarily attributed to Vitamin C as a cofactor for collagen maturation. Here, we demonstrate that Vitamin C epigenetically orchestrates osteogenic differentiation and function by modulating chromatin accessibility and priming transcriptional activity. Vitamin C regulates histone demethylation (H3K9me3 and H3K27me3) and promotes TET-mediated 5hmC DNA hydroxymethylation at promoters, enhancers and super-enhancers near bone-specific genes. This epigenetic circuit licenses osteoblastogenesis by permitting the expression of all major pro-osteogenic genes. Osteogenic cell differentiation is strictly and continuously dependent on Vitamin C, whereas Vitamin C is dispensable for adipogenesis. Importantly, deletion of 5hmC-writers, *Tet1* and *Tet2*, in Vitamin C-sufficient murine bone causes severe skeletal defects which mimic bone phenotypes of Vitamin C-insufficient *Gulo* knockout mice, a model of Vitamin C deficiency and scurvy. Thus, Vitamin C's epigenetic functions are central to osteoblastogenesis and bone formation and may be leveraged to prevent common bone-degenerating conditions.

Bone fractures due to low bone mass are a serious health concern and pose a significant burden on health care systems. In the United States, almost half (43.1%) of all adults aged 50+ years suffered from low bone mass at the femur neck, lumbar spine or both in 2017–2018[1]. This translates to an estimated 10.2 million people aged 50 and over that were affected by the bone degenerative disease osteoporosis and ~43.3 million additional people with low bone density (osteopenia) in 2010[2,3]. These individuals are at risk for bone fractures[2]. While this interconnection of low bone mass and successive fracture risks has been well-established, the prevalence of osteopenia and osteoporosis in 50+ year-olds did not improve since 2007/2008 in the United States[1]. Comprehensively understanding bone homeostasis and its deterioration on a physiological and molecular-mechanistic level will

be critical to help sustain bone health and counteract bone-debilitating conditions.

One vital component for maintaining bone health is a sustained and adequate dietary intake of micronutrients including vitamins A, C, and D3 (retinoic acid, ascorbate/L-hexuronic acid, and 1α,25-dihydroxycholecalciferol, respectively)[4–8]. Vitamin C (VitC) is produced from glucose via the enzyme L-gulono-lactone oxidase (GULO) in mice and other mammals. However, the human *GULOP* gene is nonfunctional and human health depends on dietary VitC supply. Severe VitC deficiency leads to scurvy, a serious disease characterized by critical deterioration of connective tissues accompanied by dermal, vascular and skeletal symptoms including tooth loss, skeletal fragility and bone fractures[9–12]. Although scurvy is at present considered a

relatively rare nutritional disorder, clinically confirmed cases continue to be reported globally including in Western countries[13–17]. Further, VitC deficiency remains clinically relevant in many diverse, global settings including pediatric health care[18,19], in individuals who are institutionalized (e.g., boarding school children, elderly people, priests or prisoners)[17,20–23], exposed to environmental pollution or tobacco smoke[24–28], or are part of socio-economic groups with unbalanced food diets[21,29,30]. Of note, patients with increased bone fracture risk suffer from VitC deficiency[5,6]. In turn, lower VitC intake correlates with increased risk for hip and osteoporotic fractures. Also, higher dietary VitC uptake positively correlates with bone mineral density (BMD) and reduced fracture risk[5–7,12,31]. Importantly, post-menopausal women on estrogen replacement therapy who received VitC supplementation had augmented BMD compared to estrogen only recepients[32], suggesting that VitC elicits pro-osteogenic functions that are distinct from the actions of estrogen during bone loss conditions.

Biochemically, VitC is critical for the generation and homeostasis of collagen-containing connective and mineralizing tissues (e.g., bone and teeth), as it is an essential co-factor for prolyl hydroxylating enzymes that support maturation of the collagen triple helix[33]. However, more recently it has been demonstrated that VitC can stimulate the enzymatic function of other ferrous iron- and α-ketoglutarate (αKG)-dependent dioxygenase (αKGDDs) family members including histone H3 lysine 9 (H3K9) and lysine 27 (H3K27) demethylases, as well as cytosine dioxygenases[34–36]. These epigenetic hydroxylases demethylate transcriptionally repressive H3K9me3, H3K27me3 marks or convert 5-methyl-cytosine (5mC) to 5-hydroxy-methyl-cytosine (5hmC) residues. These processes generally support chromatin opening and activation of gene expression as has been demonstrated in embryonic stem cells and other cell types[34–38]. Recent studies have demonstrated that the genomic 5hmC landscape shows tissue-specific patterns[39,40] and that 5hmC is required for the differentiation of multiple cell types including the neuronal lineage[41], the myeloid lineage[42], intestinal lineage[43] and others. For many decades, the requirement for collagen fiber assembly has been considered the main role of VitC in bone tissue, yet its recently emerging epigenetic actions raise the critical question whether VitC supports osteogenesis epigenetically beyond collagen maturation.

Here, we show that VitC epigenetically orchestrates bone formation and osteogenic differentiation by establishing a transcriptionally permissive state that selectively allows for the expression of all major bone genes. We find that VitC stimulates the activity of DNA-cytosine dioxygenases of the TET family, as well as H3K9me3 and H3K27me3 demethylases including KDM4 and KDM6 family members. The resulting epigenetic landscape consists of the activating 5hmC modification and de-repressed H3K9/H3K27 histone marks. These signatures are located at proximal and distal regions (e.g., super enhancer-like elements) near a broad spectrum of bone-selective genes, including osteogenic transcription factors, collagens, collagen-crosslinking enzymes and non-collagenous extracellular matrix (ECM) proteins that support mineralization. Bone cells during all stages of osteogenic differentiation require VitC, while it is dispensable for adipogenesis. Our study shows that VitC is an essential nutrient with epigenetic functions that precede collagen deposition and maturation and that comprehensively orchestrate osteogenic differentiation. We provide deep mechanistic insights into how VitC sustains bone health and prevents loss of bone mass.

## Results

### VitC deficiency robustly alters the transcriptional profile of bone

We examined the potential epigenetic implications of VitC in bone using a VitC-dependent mouse model, *Gulo* knockout mice. Similar to humans, these mice can no longer synthesize VitC and depend on exogenous administration of VitC through their diet[44]. *Gulo*[−/−] mice were allowed to mature normally in the presence of VitC until 15 weeks of age, at which stage VitC was withdrawn for 5 more weeks. These VitC-deficient mice exhibit severe defects in bone structure and mechanical strength compared to VitC-supplemented *Gulo* knockout mice or VitC-sufficient wild type mice (Supplementary Fig. 1a–e). Molecular phenotyping of femoral bone tissues using next-generation RNA-sequencing (RNA-seq) combined with principal component analysis (PCA) and hierarchical clustering revealed that VitC deficiency also broadly alters gene expression (Fig. 1a–d). Bone- and fat tissue-selective genes (Bone60 and Fat76 gene-sets) that we identified via the analysis of 47 human and 59 mouse RNA-seq datasets covering a broad range of tissues (Supplementary Fig. 1f) show strong down-regulation of many bone-selective genes including *Bglap*, *Dkk1* and *Dmp1* upon VitC withdrawal (Fig. 1e–h). Conversely, fat tissue-selective genes remain unchanged or increased their expression (Supplementary Fig. 2). These data demonstrate that VitC insufficiency comprehensively alters the bone transcriptional program and that VitC may affect bone beyond collagen deposition and maturation.

### VitC-controlled epigenetic programs orchestrate osteogenic gene expression

To test if the bone-related transcriptional modulations are due to VitC-dependent epigenetic adaptations, we assessed levels of two main chromatin accessibility gatekeepers, H3K9me3 and H3K27me3, as well as levels of the transcriptional activation mark 5hmC on DNA, all of which are modulated by VitC-dependent enzymes[34,45,46]. The five week long VitC depletion leads to increased H3K9me3 and H3K27me3 marks in bone when compared to VitC supplemented *Gulo*[−/−] mice (Fig. 2a, b). This effect is also apparent in soft tissues such as liver and heart. At the DNA level, bone of VitC-deficient mice exhibits the most prominent reduction of 5hmC indicating that DNA hydroxymethylation is particularly sensitive to VitC supplementation in this tissue (Fig. 2c–e). Indeed, in VitC-exposed femoral bone, 5hmC-DNA immunoprecipitation-sequencing (hMeDIP-seq) shows that cis-regulatory regions of bone-selective genes are enriched in 5hmC as demonstrated by increased peak numbers and 5hmC signal intensity (Fig. 3a, b). Conversely, VitC deprivation strongly decreases 5hmC signal near these genes. VitC-dependent 5hmC loss significantly correlates with dramatically reduced gene expression of bone-selective genes (Fig. 3c). Consistent with previous studies, 5hmC is often enriched within genes and at distal enhancers[47], which may include super-enhancers (SE) that were previously recognized as key epigenetic hubs to determine cell identity and properties[48]. To test whether VitC-dependent 5hmC may form SE-like clusters, we performed SE analyses on our hMeDIP-seq data. Strikingly, almost half of the Bone60 genes have a VitC-dependent 5hmC SE-like cluster in their vicinity (Fig. 3d) and VitC withdrawal reduces 5hmC SE signal in the majority of these SEs (Fig. 3e, f). These data indicate that the robust VitC-dependent transcriptional changes in bone are orchestrated by epigenetic programs that act on different genomic regulatory elements to drive the osteogenic program.

### VitC is required throughout osteogenic differentiation but is dispensable for adipogenic differentiation of mouse and human progenitors

To probe for the underlying phenotypic VitC-dependent mechanisms, we assessed osteogenic lineage commitment of primary bone marrow stromal cells (BMSCs) extracted from *Gulo*[−/−] mice. BMSCs from VitC-depleted *Gulo*[−/−] mice contain fewer but larger colony forming units compared to BMSCs from VitC-supplemented *Gulo*[−/−] mice (Fig. 4a). Furthermore, BMSCs lacking VitC in vitro are incapable of ECM deposition and mineralization despite the presence of β-glycerophosphate as external phosphate source to promote mineral deposition (Fig. 4b, c). In contrast, VitC administration to BMSCs from previously VitC-deprived mice restores their osteogenic potential (Fig. 4b, c). Reminiscent of data in femur tissue, H3K9me3 and H3K27me3 demethylation as well as 5hmC levels tightly correlate with VitC administration and osteogenic differentiation (Fig. 4d). Of note,

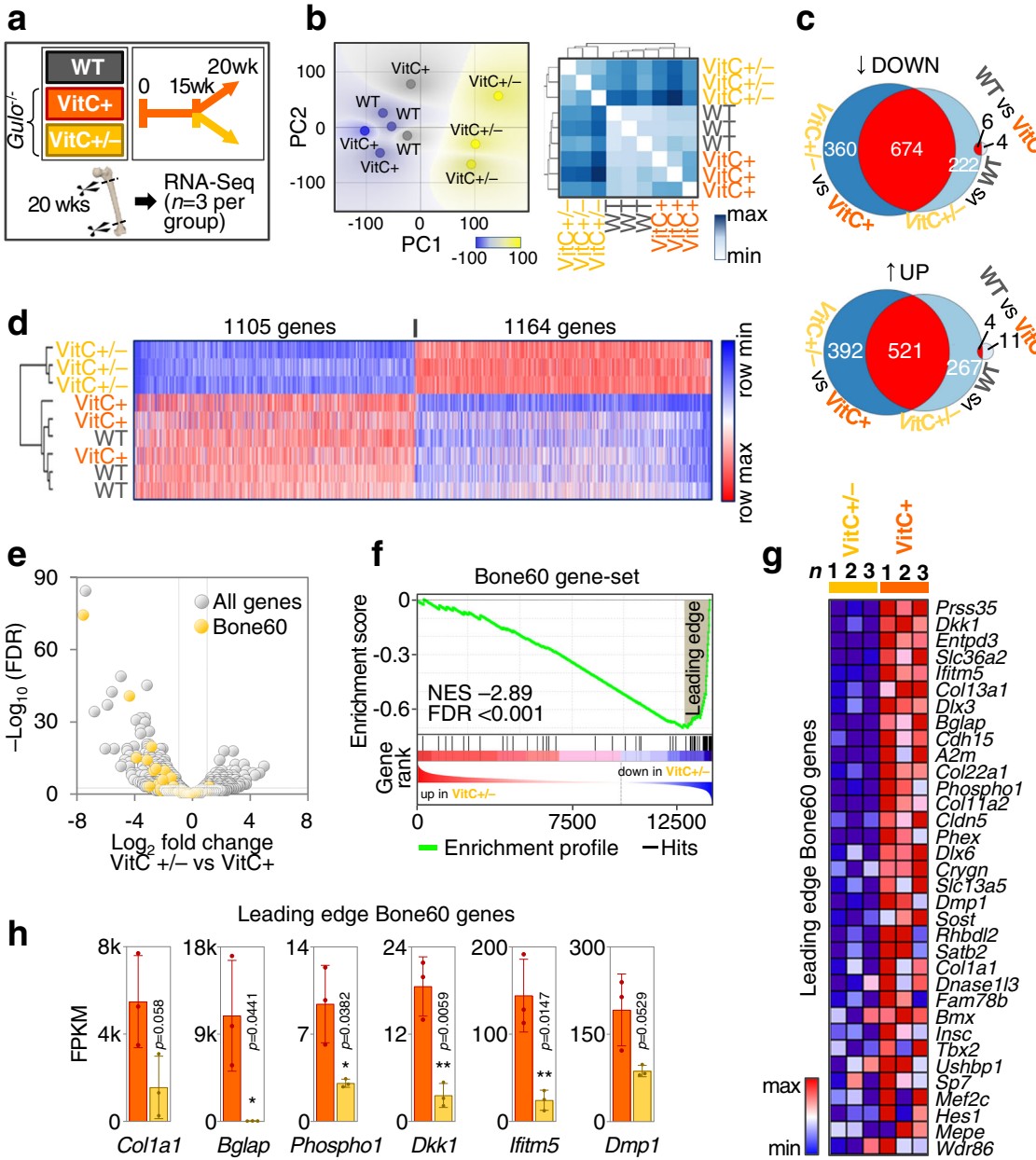

**Fig. 1 | Vitamin C deficiency causes vast transcriptional changes in bone.**
**a** Experimental setup; wk, week. **b** RNA-seq analysis of femoral bone using principal component analysis and hierarchical clustering. **c** Venn diagrams depicting differentially expressed genes (*p*-adj < 0.05). **d** Gene expression heatmap displaying differentially expressed genes between VitC-positive and VitC-negative mice (*p*-adj < 0.05). **e**–**h** Volcano plot and GSEA analysis of the Bone60 gene-set in VitC ± versus VitC+ RNA-seq data, NES normalized enrichment score, FDR false discovery rate. **g** Heatmap and **h** gene expression bar charts depict leading-edge genes. Bar graphs represent mean ± SD. *$p < 0.05$; **$p < 0.01$; FDR adjusted Wald test (**e**), FDR-adjusted two-tailed, unpaired *t* tests; *N* = 3 per group from biologically independent animals (**h**). Source data are provided as a Source Data File.

exceptionally high 5hmC levels are evident in mature osteoblasts that produce mineralization nodules (Fig. 4e). Even after day 35 of differentiation, i.e., at a late stage of osteogenic lineage maturation when cells are already embedded in a mineralized ECM and express the early osteocyte marker *Dmp1*, continuous VitC supplementation is necessary to allow further differentiation, mineralization and expression of well-established osteoblast and osteocyte markers including *Phospho1* and *Sost* (Fig. 4f, g). Notably, VitC does not promote BMSC differentiation into the adipogenic lineage as measured by the expression of key adipogenic markers and Red Oil O staining (Supplementary Fig. 3a), further demonstrating that the role of VitC in BMSC differentiation is selective for the osteogenic lineage. Despite the adipogenic conditions, VitC addition promotes the expression of osteogenic markers (Supplementary Fig. 3b). Correlating with this pro-osteogenic signature during adipogenic differentiation, the overall levels of H3K9me3 and H3K27me3 decrease, while 5hmC levels increase after VitC administration (Supplementary Fig. 3c, d). Remarkably, the highest 5hmC levels are seen in clusters of cells expressing the osteogenic marker BGLAP (Supplementary Fig. 3e).

To assess whether VitC's role in osteogenic differentiation and function is conserved in humans, we studied BMSCs derived from healthy human donors. Here, ECM deposition and mineralization, ALPL activity and osteogenic marker gene expression are only induced after VitC addition (Supplementary Fig. 4a–c). Similarly, H3K9me3 and H3K27me3 demethylation and 5hmC gain is only observed in the presence of VitC (Supplementary Fig. 4d, e). Reminiscent to BMSCs

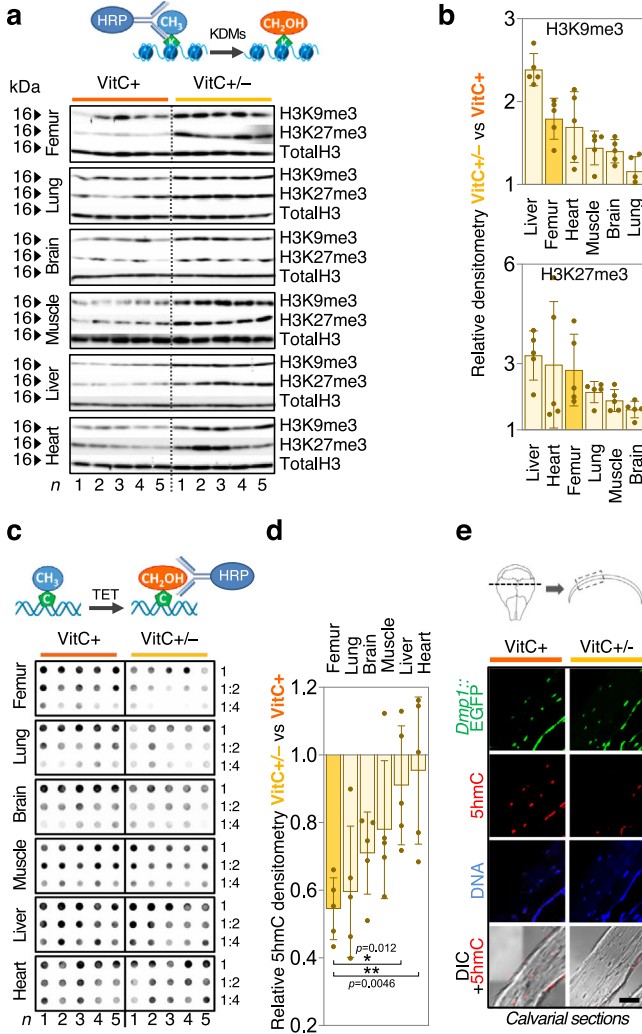

**Fig. 2 | Histone and DNA demethylation signatures in bone are highly sensitive to Vitamin C. a** H3K9me3 and H3K27me3 western blots in indicated tissues from 20 week old *Gulo*$^{-/-}$ mice supplemented with VitC (VitC+) or without VitC from WK 15 to WK 20 (VitC ±). **b** Relative western blot quantitation to (**a**) shown as fold change to VitC+. **c** 5hmC dot blot in indicated tissues from same experimental setup as in **a** and relative quantitation (**d**). **e** Immunofluorescence of 5hmC in cal-varial sections; *Gulo*$^{-/-}$ mice were crossed with transgenic *Dmp1*-EGFP$_{Topaz}$ mice to visualize osteocytes; DIC, differential interference contrast, scale bar represents 50 µm. Bar graph represent mean ± SD; *$p < 0.05$; **$p < 0.01$; one-way ANOVA with Dunnett's multiple comparison tests (**d**); $N = 5$ per tissue and group from biologically independent animals (**b, d**). Source data are provided as a Source Data File.

derived from *Gulo*$^{-/-}$ mice, adipogenic differentiation does not require VitC (Supplementary Fig. 5a, b). Decreases in H3K9me3 and H3K27me3 demethylation and increased 5hmC levels are apparent upon VitC addition, and correlate with upregulation of osteogenic markers during adipogenic differentiation (Supplementary Fig. 5c–f). Collectively, these data suggest that VitC is sufficient to selectively prime BMSCs towards osteogenesis irrespective of species and that adipogenesis is orchestrated by different epigenetic patterns and mechanisms[49].

**VitC promotes osteogenic transcriptional programs independent of collagen synthesis during osteoblastogenesis**
Next, we investigated the relationship between the gene regulatory effects of VitC and its role as a cofactor of collagen hydroxylases that are required to generate a collagenous ECM. VitC rapidly stimulates the expression of collagen hydroxylases and crosslinker genes in BMSCs differentiated towards the osteogenic lineage (Fig. 5a). To test

whether the collagenous matrix rather than the epigenetic role of VitC supports osteogenic gene expression, we inhibited collagen deposition in the presence of VitC during osteogenic differentiation of *Gulo*$^{-/-}$ BMSCs (Fig. 5b, c). As expected, treatment with the prolyl hydroxylase inhibitor 1,4-DPCA efficiently prevents ECM deposition and does not permit matrix mineralization (Fig. 5d, e). Yet, regardless of collagenous matrix deficiency, VitC is still able to induce the expression of mature osteoblastic genes such as *Bglap2, Dlx3* or *Col22a1* and to a lesser extent of some osteocyte markers (Fig. 5f). Similarly, human BMSCs co-treated with 1,4-DPCA and VitC retained the osteogenic gene expression profile despite the absence of a collagenous ECM (Fig. 5g, h). We conclude that VitC is largely sufficient to conduct the osteogenic transcriptional program, independent of the concurrent deposition of a collagenous matrix.

**VitC promotes osteogenic differentiation independent of its role as an antioxidant**
VitC has also a well-established role in controlling the cellular redox balance as it can act as an antioxidant to neutralize a variety of reactive oxygen species (ROS)[50,51]. To test whether antioxidative proprieties play an essential role in promoting osteogenic differentiation of BMSCs, we compared VitC to an alternative antioxidant, N-Acetyl Cysteine (NAC). Unlike VitC, NAC addition does not induce osteogenic differentiation nor affect H3K9me3/H3K27me3 demethylation or 5hmC levels (Supplementary Fig. 6) suggesting that VitC promotes osteogenic differentiation independent of its role as free radical scavenger. Furthermore, VitC has been shown to potentiate reprogramming efficiency of induced pluripotent stem cells in part by alleviating ROS and counteracting cellular senescence, a cell fate characterized by a durable cell cycle arrest and a bioactive secretome[52,53]. We found that VitC does not alter induction of cellular senescence of BMSCs as assessed by the activity of senescence-associated β-galactosidase, proliferation and expression of senescence markers, *p16, p19* or *p21* (Supplementary Fig. 7a–c). Unlike pluripotent mouse embryonic stem cells (mESCs), BMSCs are a heterologous collection of bone marrow stromal cells with limited potency and self-renewal capacity[54,55]. Consistent with this, when comparing RNA-seq data of mESCs and BMSCs we found that these cell types show clear overall transcriptomic divergence (Supplementary Fig. 7d). BMSCs hardly express classical stem cell markers such as *Oct4, Sox2* or *Klf4* (Supplementary Fig. 7e). Additionally, temporal gene expression patterns of BMSC markers[56] are largely unaffected during VitC-mediated osteogenic differentiation (Supplementary Fig. 7f). Collectively, these data indicate that the pro-osteogenic properties of VitC are unlikely due to modulated oxidative stress or retained stemness.

Importantly, VitC is an essential cofactor for many ferrous iron- and α-ketoglutarate (αKG)-dependent epigenetic dioxygenases (αKGDDs). Family members include Jumonji-C domain-containing histone demethylases (JHDMs/KDMs) as well as the ten-eleven translocation (TET) family of DNA hydroxylases[57]. The catalytic activity of these enzymes depends on the reduction of Fe$^{3+}$ to Fe$^{2+}$ by VitC[35,58,59]. To test whether the epigenetic role of VitC during osteogenesis requires this catalytic reaction, we utilized the competitive inhibitor L-2-hydroxyglutarate (L-2-HG) to suppress the catalytic activity of αKGDDs. Strikingly, L-2-HG treatment almost completely ablates VitC-dependent induction of 5hmC and suppresses VitC induced reduction in H3K9me3/H3K27me3 levels during osteogenic differentiation of mouse BMSCs and pre-osteoblasts (Fig. 6). These results suggest that during osteogenesis, a main function of VitC is to catalyze the epigenetic activity of αKGDDs in order to orchestrate osteogenic differentiation.

**Proximal and distal 5hmC levels near osteogenic loci correlate with bone gene expression and bone-related phenotypes**
To examine the underlying molecular mechanism by which VitC-dependent αKGDD(s) direct bone specific gene expression during

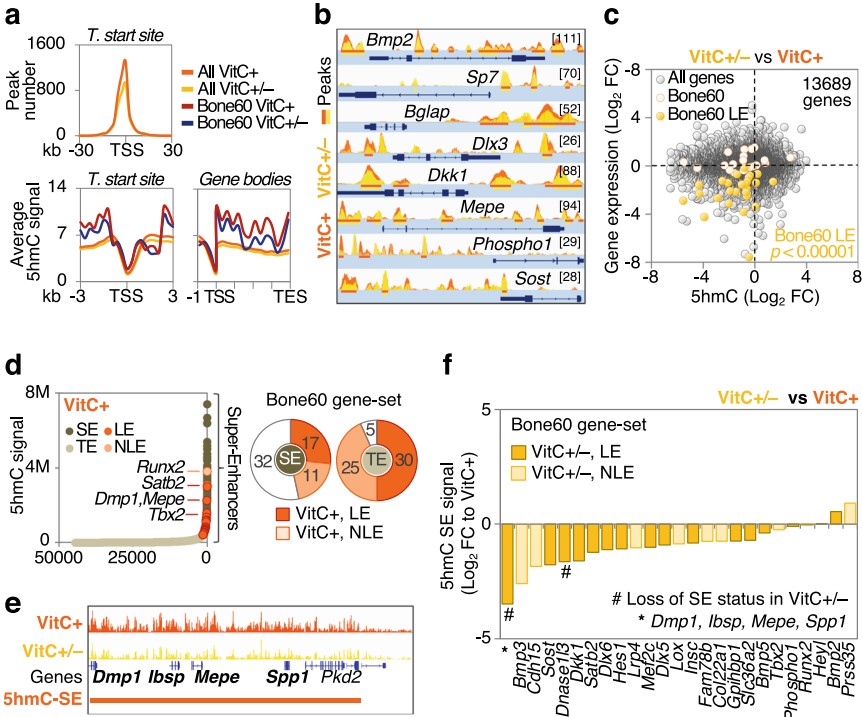

**Fig. 3 | Vitamin C-sensitive DNA hydroxymethylation levels distal and proximal to bone-specific genes correlate with their expression. a** hMeDIP-seq analysis of femoral bone showing average 5hmC peak number around transcriptional start sites (TSS), average 5hmC signal around TSS or in gene bodies for all genes or for the Bone60 gene-set; TES, transcriptional end site. **b** Overlaid 5hmC peak occupancy comparing VitC+ and VitC ± groups near Bone60 genes in femurs; [value] =max peak scale. **c** Correlation between gene expression and 5hmC occupancy (TSS +/−30 kb) in VitC +/− vs VitC+ femurs. LE, leading edge as per GSEA, FC, fold change. **d** Super-enhancer (SE) based clustering of 5hmC peaks with *Runx2* as the highest ranking Bone60 gene and number of Bone60 genes associated with 5hmC-SE and 5hmC-TE as well as example of a 5hmC-SE (**e**); TE, typical enhancers; NLE, non-leading edge. **f** Ranked decrease in 5hmC signal at 5hmC-SE after VitC withdrawal. Two-sided Fishers exact test (**c**). Source data are provided as a Source Data File.

osteoblastogenesis, we employed MC3T3-E1 pre-osteoblasts as a model system in vitro. First, we validated that VitC administration alters epigenetic marks at both the histone and DNA levels and stimulates osteoblast maturation in these cells (Supplementary Fig. 8a–d). Similar to bone tissue, VitC-dependent 5hmC levels robustly increase near promoters and within gene bodies of bone-related genes and positively correlated with their mRNA expression (Supplementary Fig. 8e–g). VitC-dependent 5hmC peaks again form SE-like clusters in the vicinity of bone related genes, which overlap with H3K27ac super-enhancers found in BMSCs at advanced stages of osteogenic differentiation (Supplementary Fig. 9). H3K9me3 or H3K27me3 ChIP-seq experiments revealed that VitC supplementation causes loss of H3K27me3 around transcription start sites and gene bodies of bone-specific genes (Supplementary Fig. 10a). However, unlike for 5hmC, overall bone-specific gene expression does not correlate with altered H3K9me3 or H3K27me3 levels around these genes and their promoters (Supplementary Fig. 10b, c). In fact, only a subset of bone-specific genes is expressed in a VitC-dependent, H3K9me3-dependent fashion, such as *Phospho1* and *Dmp1*, or in a VitC-dependent, H3K27me3-dependent fashion including *Siglec15*, *Bmp8b* and *Notch3* (Supplementary Fig. 10b, d). Closer examination of the genomic distribution of these epigenetic marks shows that a large proportion (37–70%) of H3K9me3, H3K27me3 or 5hmC peaks is located in distal intergenic regions (Supplementary Fig 11a). Thus, we performed GREAT analyses on distal loci that exhibit the most significant alterations in peak signals upon VitC deficiency. Only sites exhibiting 5hmC loss strongly correlate with a variety of bone-associated phenotypes, while H3K27me3 occupied loci correlate to a significant extent with neurological phenotypes (Supplementary Fig. 11b). No phenotypes could be linked to altered distal H3K9me3 related loci. Thus, VitC directly controls osteoblastogenesis of MC3T3-E1 cells by installing a pro-osteogenic, transcriptionally permissive epigenetic state at bone-specific genes and bone relevant distal intergenic loci that is primarily driven by 5hmC.

## 5mC hydroxylase TET2 is particularly involved in the osteogenic program

Next, we assessed which VitC-dependent αKGDD(s) may contribute to osteoblast differentiation by systematically examining established VitC-sensitive H3K9me3 or H3K27me3 demethylases (*Kdm4a*, *Kdm4b*, *Kdm4c*, or *Kdm6a*, *Kdm6b*, *Jhdm1d* respectively) and 5mC hydroxylases (*Tet1*, *Tet2*, *Tet3*). Most of these genes are well-expressed in MC3T3-E1 pre-osteoblasts, MLO-A5 pre-osteocytes and mouse femurs (Fig. 7a, Supplementary Fig. 12a). We subsequently performed stable shRNA-mediated depletion experiments of these enzymes during VitC-mediated osteoblastic differentiation. We also included co-depletion conditions to assess the combined functions of epigenetic modulators within the same class and to account for potential compensation among family members (Fig. 7b–d, Supplementary Fig. 12b–d).

*Tet2* depletion in MC3T3-E1 cells causes the strongest decrease in ECM deposition, collagen cross linking, mineralization and osteogenic gene expression (Fig. 7c, d). *Tet2* is among the most important epigenetic enzymes for mineralization of MLO-A5 cell cultures as well (Supplementary Fig. 12c, d). *Tet1* is also critical for collagen cross linking and mineralization, but it is largely dispensable for ECM deposition and expression of the measured osteogenic markers, whereas *Tet3* appears to counteract osteoblastogenesis of MC3T3-E1 and MLO-A5 cells (Fig. 7c, d, Supplementary Fig. 12c, d). Accordingly, VitC-induced 5hmC accumulation is to a great extent dependent on *Tet2* in MC3T3-E1 osteoblasts (Fig. 7e). These results indicate that the VitC-dependent TET hydroxylases have distinct roles during

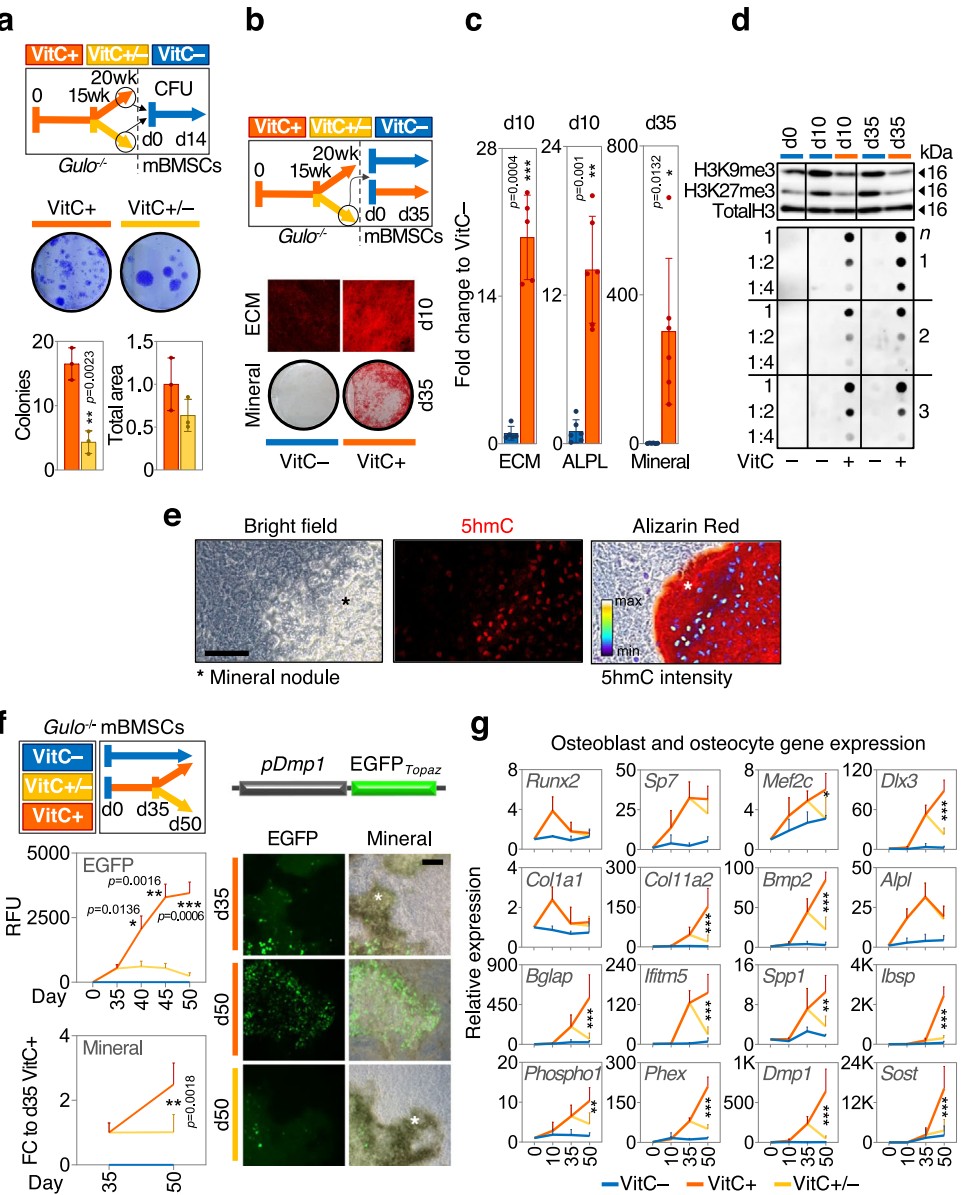

**Fig. 4 | Vitamin C governs all differentiation stages of the osteogenic lineage.**
**a** Colony formation unit assay (CFU) of BMSCs isolated from VitC treated (VitC+) or VitC depleted (VitC ±) *Gulo*⁻/⁻ mice from week 15 to week 20. **b** Extracellular matrix (ECM) deposition and ECM mineralization in BMSCs isolated from VitC-deficient *Gulo*⁻/⁻ mice. **c** Quantitation of ECM deposition, alkaline phosphatase activity (ALPL) and ECM mineralization in VitC treated and untreated BMSCs. **d** Western blot for H3K9me3 and H3K27me3 and dot blot for 5hmC during osteogenic lineage progression. **e** 5hmC immunofluorescence and alizarin red stain in mature osteoblasts within mineralizing nodules (*). **f** Quantification of *Dmp1* promoter activity (EGFP fluorescent signal) and ECM mineralization in cultures in which VitC was removed after D35 and representative images from experiments quantified in the left panel. (*) mineralized nodules; BMSCs from *Gulo*⁻/⁻ mice with Dmp1-EGFP_Topaz reporter

were used for this experiment; RFU, relative fluorescence units. **g** mRNA expression of osteoblast and osteocyte markers in BMSCs with or without VitC, or cultures in which VitC was removed starting at D35 (VitC +/−). Scale bars in (**e**) and (**f**) represent 50 μm. Graphs represent mean ± SD, *$p < 0.05$; **$p < 0.01$; ***$p < 0.001$. Unpaired two-tailed *t* test in (**a**), paired two-tailed *t* test in (**c**), two-way ANOVA with Tukey's (EGFP) and Sidak's (Mineral) multiple comparison tests in (**f**), two-way ANOVA with Tukey's multiple comparison tests in (**g**); significances in (**g**) represent D50 VitC+ vs D50 VitC ±. *N* = 3 (**a**, **f** mineral, **g**), *n* = 4 (**f**, EGFP), *n* = 5 (**c**, ECM) *n* = 6 (**c**, ALPL & Mineral) per group from cells derived from biologically independent animals. Source data as well as exact *p* = values for all comparisons in (**g**) are provided in the Source Data File.

osteogenesis and that 5hmC deposition by TET2 is vital for osteogenic differentiation and function.

We further found that knock down of H3K9me3 demethylases *Kdm4a* and *Kdm4b* mimic *Tet2* loss except for their key role towards ALPL activity while *Kdm6a* and *Kdm6b* are important for late stages of differentiation in MC3T3-E1 cells as well as MLO-A5 pre-osteocytes. Interestingly, co-depletion of at least two H3K27me demethylase family members (e.g., *Kdm6a* and *Kdm6b*) potentiates the deleterious effect on some parameters during osteoblastic differentiation

suggesting that these enzymes are redundant and may compensate to some extent during osteogenesis.

We validated these results by knocking down all candidate epigenetic modulators in primary mouse BMSCs (Supplementary Fig. 13). In these cells *Tet2* is also the most critical 5mC hydroxylase during osteogenic differentiation as determined by strong reductions in ECM deposition and mineralization as well as ALPL activity (Supplementary Fig. 13b). Similarly, depletion of H3K9me3 or H3K27me3 demethylases largely mimics the effects seen in MC3T3-E1 and MLO-A5 cells.

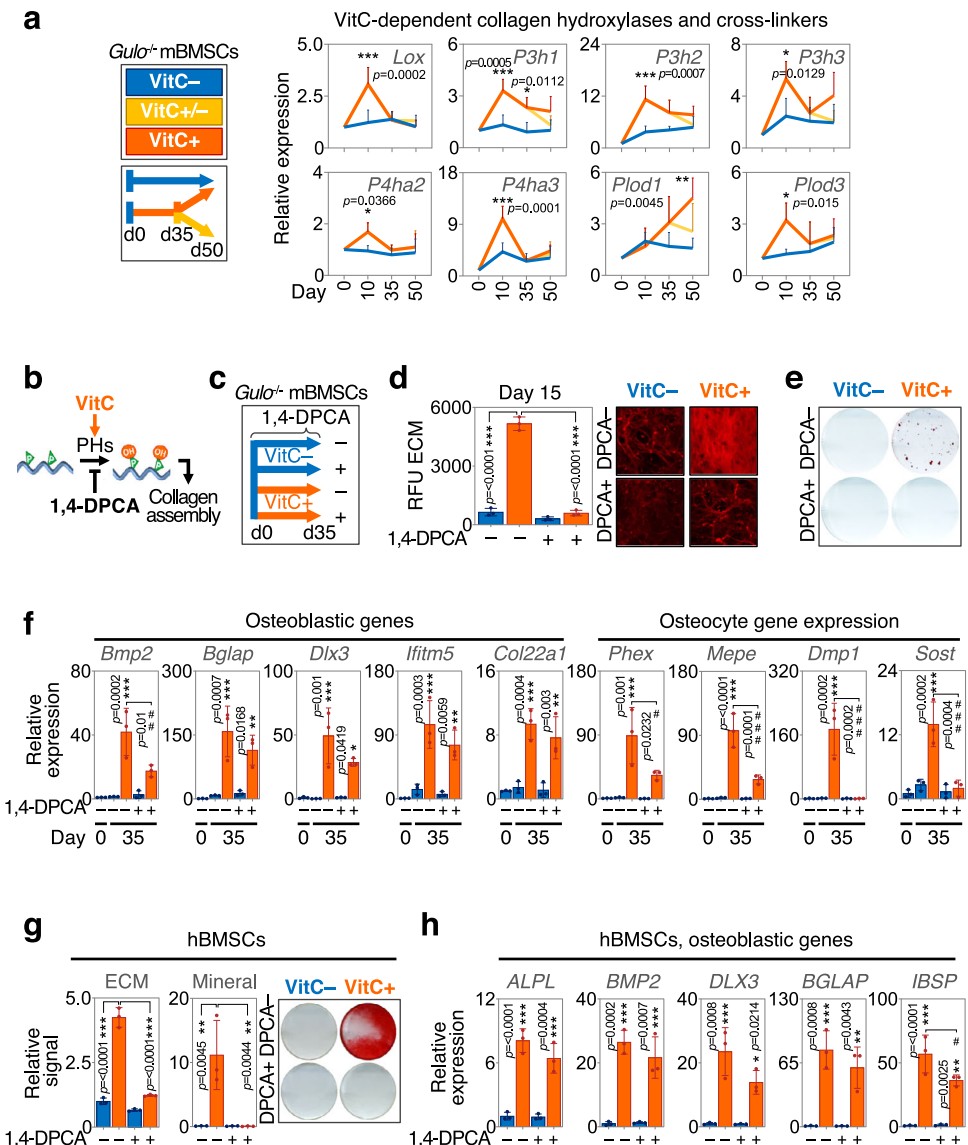

**Fig. 5 | Vitamin C-dependent collagenous matrix formation is dispensable for Vitamin C-mediated osteoblastic gene expression. a** Gene expression of collagen hydroxylases and cross-linkers in BMCSs from 5-week VitC-deficient *Gulo⁻ᐟ⁻* mice cultured in the presence or absence of VitC, or treated with VitC until day 35 after which VitC was removed. **b** Inhibition of VitC-dependent extracellular matrix (ECM) deposition by 1,4-DPCA. **c** Experimental setup using 1,4-DPCA during VitC-induced osteogenic differentiation. **d–f** Effects of 1,4-DPCA on ECM deposition (RFU, relative fluorescence units) (**d**), ECM mineralization (**e**) and osteogenic gene expression (**f**); VitC-untreated BMSCs at day 0 were used as an additional control. **g** Effects of 1,4-DPCA on ECM deposition and ECM mineralization in human BMSCs during osteogenic differentiation. **h** Osteogenic gene expression in hBMSCs treated with or without 1,4-DPCA. Graphs represent mean ± SD, *$p < 0.05$; **$p < 0.01$; ***$p < 0.001$ comparing VitC+ and VitC- groups at corresponding time points (**a**) test against VitC- D0 (**f**) and test against VitC-/DPCA- if not otherwise noted (**g**, **h**). #$p < 0.05$; ##$p < 0.01$; ###$p < 0.001$ comparing VitC+ groups with vs without 1,4-DPCA at D35 (**f**, **h**); in **h** for *IBSP* # $p = 0.0447$. Two-way ANOVA (**a**) and one-way ANOVA (**d**, **f**, **g**, **h**) with Tukey's multiple comparison tests. $N = 3$ (**a**, **d**, **f–h**) per group from cells derived from biologically independent animals/donors. Source data are provided as a Source Data File.

Complementary dCas9-VP64/CRISPR activation-mediated over-expression analyses of all candidates verified that TET2 activation provokes VitC-dependent 5hmC accumulation and promotes MC3T3-E1 osteoblast differentiation (Fig. 8). In contrast, CRISPR-mediated activation of H3K9me3 or H3K27me3 demethylases only has minor consequences for MC3T3-E1 osteoblastogenesis (Supplementary Fig. 14), with only the overexpression of KDM4C or KDM6A causing some enhancement in osteoblastic differentiation. Next, we tested whether the pro-osteogenic effect of TET2 could be potentiated by co-expressing KDM4C or KDM6A and vice versa (Supplementary Fig. 15). However, we found that none of the combinatorial conditions surpasses the pro-osteogenic properties of individual genes (Supplementary Fig. 16). Collectively, these molecular profiling studies

strengthen earlier conclusions that TET-enzymes, in particular TET2, are central epigenetic enzymes orchestrating diverse features of osteoblastogenesis.

## 5hmC generation rather than 5mC loss is vital for osteoblastic differentiation

To investigate whether opening of chromatin by removal of the repressive 5mC mark during the active DNA demethylation cycle rather than the establishment of the activating 5hmC residue is responsible for the effects of VitC on osteogenic gene expression, we performed experiments in MC3T3-E1 osteoblasts using the DNA methyltransferase inhibitor RG108 (Fig. 9a). As expected, RG108 reduces 5mC levels in the absence and presence of VitC (Fig. 9b).

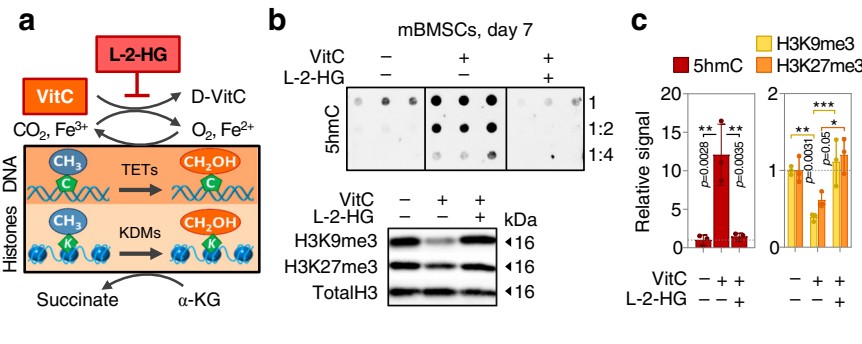

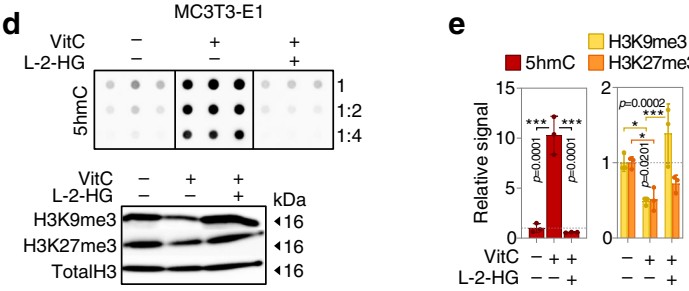

**Fig. 6 | Vitamin C controls the activity of epigenetic αKGDDs during osteogenic differentiation. a** Inhibition of VitC dependent α-ketoglutarate dehydrogenases by L-2-hydroxyglutarate (L-2-HG) during osteogenic differentiation. **b** Inhibition of VitC dependent 5hmC synthesis and H3K9me3 as well as H3K27me3 demethylation by L-2-HG during osteogenic differentiation of primary mBMSCs and quantification of blots (**c**). **d** Effects of L-2-HG in differentiating calvarial MC3T3-E1 pre-osteoblasts on 5hmC deposition and H3K9me3 as well as H3K27me3 demethylation and

quantification of the blots (**e**). Bar graphs represent mean ± SD, *p < 0.05; **p < 0.01; ***p < 0.001. One-way ANOVA with Tukey's multiple comparison test for dot blot quantifications and two-way ANOVA analysis with Tukey's multiple comparison tests for western blot quantifications in **c, e**. In (**c**), for H3K9me3 ***p = 0.0008, in (**e**), for H3K9me3 *p = 0.0155. N = 3 (**c, e**) per group from cells derived from biologically independent animals (**c**) or independent biological experiments (**e**). Source data are provided as a Source Data File.

However, this effect does not promote nor perturb VitC-dependent osteoblast differentiation (Fig. 9c, e). Interestingly, we also found that VitC-induced 5hmC is a stable epigenetic mark and is not further oxidized to 5-formylcytosine (5fC) during osteoblastic maturation (Fig. 9d). These findings demonstrate that 5hmC is specifically generated via VitC and represents a transcriptionally active epigenetic mark that permits and sustains osteogenic differentiation.

### TET enzymes are required for healthy bone formation

Because we found 5hmC generation to be one of the most fundamental VitC-dependent mechanisms during osteogenic differentiation, we generated mice with bone-selective conditional knockout for pro-osteogenic 5mC hydroxylases *Tet1 (Prrx1-Cre; Tet1 flox/flox)*, *Tet2 (Prrx1-Cre; Tet2 flox/flox)* or both *(Prrx1-Cre; Tet1 flox/flox; Tet2 flox/flox)* (Fig. 10a). *Prrx1-Cre*-expressing mice were used as controls. Indeed, *Tet1/2* loss significantly reduces global 5hmC levels in mouse femurs despite these mice being self-sufficient for VitC (Fig. 10c). The broadly altered 5hmC patterns between control mice and *Tet1/2* double knockouts were also confirmed by principal component analysis of femoral 5hmC MeDIP-Seq data (Fig. 10d). Decreased 5hmC levels around bone specific genes is accompanied by vastly diminished expression of bone-specific genes in *Tet1/2* double knockout mice (Fig. 10e−g). Furthermore, GREAT analysis of distal intergenic regions harboring the most dramatic 5hmC losses in conditional *Tet1/2* knockout femurs significantly correlates with multiple bone-related phenotypes including decreased bone volume or decreased trabecular bone thickness (Fig. 10h). Indeed, as shown by μCT analysis of the L5 spine, *Tet1/2* conditional knockout mice suffer from bone deficiency as shown by strongly reduced bone volume, trabecular number and trabecular thickness as well as increased trabecular separation (Fig. 10i, j). These aberrant skeletal parameters phenocopy the bone presentation of VitC deficient *Gulo⁻/⁻* mice. These observations are in line with

the notion that impaired osteogenesis due to VitC deficiency is, at least in part, due to deficiency in TET enzyme activity and impaired 5hmC generation. Thus, these results ascertain the essential role of 5hmC and VitC-dependent TET enzymes during osteogenesis and bone homeostasis.

## Discussion

VitC is a unique micronutrient with multiple biochemical activities and physiological functions required for human health. Its role as co-factor for many αKGDDs such as histone demethylases and the TET-class of cytosine hydroxylases makes VitC an epigenetic compound with broad significance in pathophysiological contexts[36,45,60,61]. The findings presented here reveal that bone integrity is critically dependent on the epigenetic actions of VitC that are required for osteoblastic differentiation and bone matrix mineralization. While VitC also controls collagen matrix deposition in bone via collagen hydroxylation, this role appears mechanistically downstream from epigenetic control (Supplementary Fig. 17).

Our functional analyses on VitC-dependent epigenetic modulators during osteogenic differentiation led to several insights into physiological bone formation and osteogenic lineage differentiation.

First, we found that perturbation of VitC-dependent epigenetic circuits involving histone H3 demethylation and 5hmC formation prevents adequate osteogenesis. We also found serious skeletal pathologies such as dramatically reduced bone mass not only in VitC-insufficient *Gulo* knockout mice, but also in VitC-sufficient *Tet1/2* conditional knockout mice which is consistent with a previous study[62]. In human studies, lower VitC intake correlates with increased bone fracture risk, while increased VitC uptake correlates with higher BMD and reduced fractures risk[7,12,31]. Therefore, we propose that VitC is a crucial epigenetically acting agent that sustains bone homeostasis and bone health in humans.

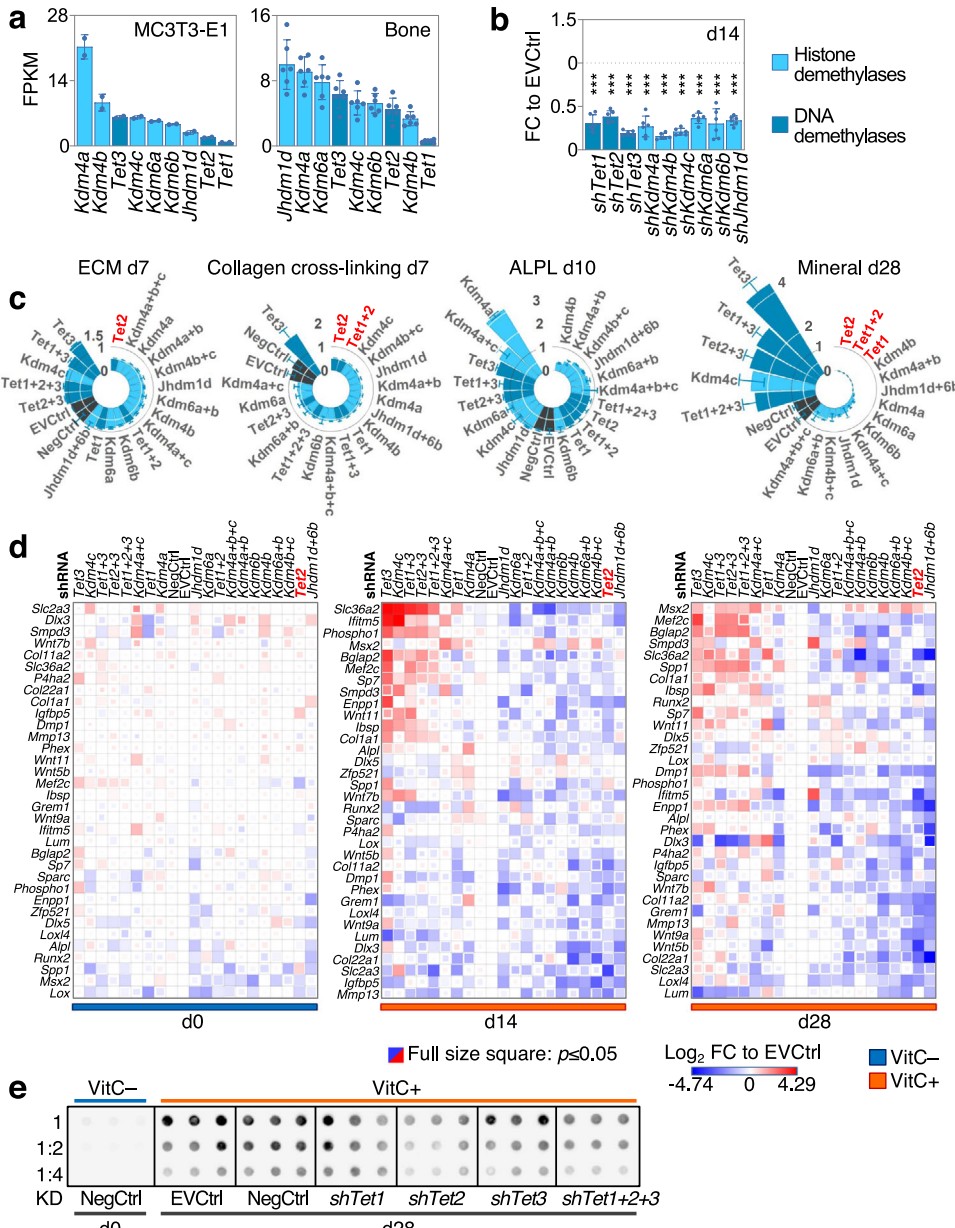

**Fig. 7 | Loss of Histone demethylases and DNA hydroxymethylases, particularly TET2, impairs osteogenic differentiation. a** Gene expression of VitC-dependent epigenetic hydroxylases in MC3T3-E1 cells and mouse femur. **b** shRNA mediated hydroxylases knockdown efficiencies in MC3T3-E1 cells. **c** Ranked relative extracellular matrix (ECM) deposition, collagen cross links measured via FTIR, alkaline phosphatase activity (ALPL) and ECM mineralization after suppression of hydroxylases in MC3T3-E1. **d** Two-way ranked osteoblastic gene expressions after hydroxylases knockdowns in MC3T3-E1 cells; FC, fold change; EV, empty vector. **e** 5hmC dot blot after *Tet* knockdowns in MC3T3-E1 cells. Graphs represent mean ± SD, ***$p < 0.0001$. One-way ANOVA with Dunnett's multiple comparison test (**b**), unpaired, two tailed *t* test (**d**). Data points in (**b**–**d**) were compared to EVCtrl. $N = 2$ (**a**, MC3T3-E1), $n = 6$ (**a**, bone), $n = 3$/tested shRNA (**b**), $n = 3$ (**c**) from biologically independent experiments (**a**, MC3T3-E1, **b**,**c**) or from biologically independent animals (**a**, bone). Source data are provided as a Source Data File.

Second, our results indicate that VitC-regulated chromatin accessibility and activation of gene expression, particularly through TET2-mediated DNA-cytosine hydroxymethylation, is a key prerequisite for osteogenic lineage commitment and progression. VitC is continuously required throughout osteogenic differentiation, from lineage commitment of BMSCs to maturation of osteoblasts and osteocytes. Depletion of VitC in maturing osteoblasts disables further osteogenic differentiation. These observations highlight the significance of adequate and consistent VitC intake through a balanced diet to support bone accrual and homeostasis. Interestingly, in line with the existing literature, we found that TET3 suppression enhances osteogenesis[63], indicating that each TET enzyme may fulfill distinct biological roles[64–66]. It is conceivable that the specificity of TET3 during

osteogenesis could differ from other TET enzymes, for example by controlling 5mC hydroxylation of non-bone-specific sites and targets genes and thus controlling different cellular functions.

Third, strikingly 5hmC marks are especially sensitive to VitC loss in bone compared to several non-osseous tissues from *Gulo* knockout mice and disproportionally high 5hmC levels are found in mature osteoblasts embedded within mineralized nodules in vitro. Our findings are also consistent with indirect evidence that 5hmC positively correlates with osteoblast differentiation in vitro[67–70]. Two recent, comprehensive studies on the genomic 5hmC landscape across multiple human tissues underline 5hmC's tissue selectivity also in humans[39,40]. Although neither study included bone tissue, the overall conclusion that 5hmC accumulates at tissue-specific regions near

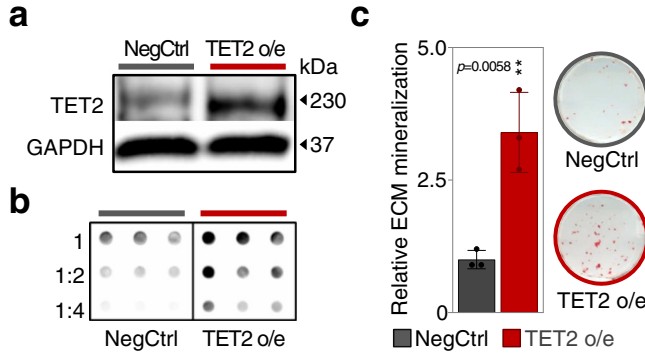

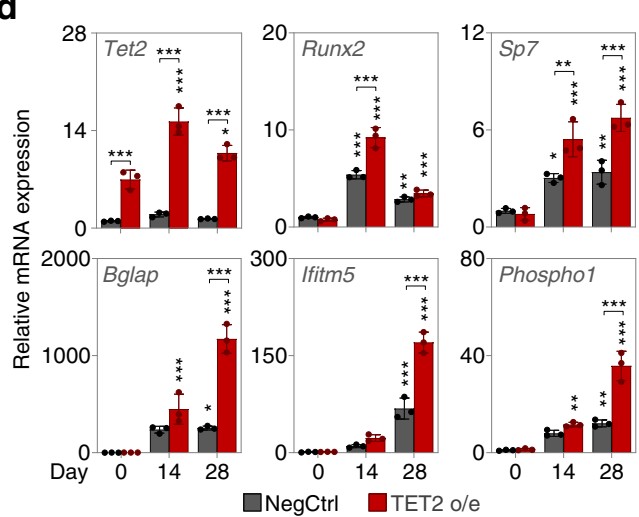

**Fig. 8 | TET2 overexpression potentiates osteoblastic differentiation. a** TET2 western blot after CRISPR-dCas9-mediated TET2 o/e (overexpression) and **b** 5hmC dot blot under the same conditions. **c** ECM mineralization after TET2 o/e at day 21 of differentiation. **d** Osteoblastic gene expression during TET2 o/e. Bar graphs represent mean ± SD; *$p < 0.05$; **$p < 0.01$; ***$p < 0.001$. Unpaired two-tailed $t$ test in (**c**); two-way ANOVA analysis with Tukey's multiple comparison tests, groups compared to D0 Ctrl or as marked in **d**. $N = 3$ (**c**, **d**) per group, from biologically independent experiments. Source data as well as exact $p =$ values for all comparisons in **d** are provided in the Source Data File.

tissue-selective genes is in full agreement with our observations. Interestingly, adipogenesis does not require VitC even though both osteogenic and adipogenic lineages can originate from the same BMSC culture. While this strict dependency of VitC and 5hmC towards osteogenesis requires further experimentations, this divergence might involve cell type-specific, mutually-exclusive characteristics such as differing metabolic properties or epigenetic requirements[49].

The link between 5hmC and advanced differentiation appears to be independent of cell division as osteoblasts stop proliferating at early stages of differentiation, concomitant with ECM production[71,72]. Supporting this observation, 5hmC induction during osteogenic differentiation appears to be largely independent of 5mC levels and activity of DNA methyltransferases (DNMTs), which typically are dependent on cell divisions[73]. We also did not find evidence of further conversion of 5hmC to 5fC during osteogenic differentiation, suggesting that 5hmC accumulation rather than active DNA demethylation per se is driving bone cell maturation. However, comprehensive detection of formylated and carboxylate cytosines (5fC and 5acC, respectively) remains technologically challenging, thus precluding a more detailed characterization of these DNA modifications and their physiological roles in bone biology at present.

Forth, while we discovered global, VitC-sensitive changes in H3K9me3/H3K27me3 and 5hmC levels in bone and osteogenic cells, VitC-dependent adaptations critical to osteogenesis are found near a specific, bone-related gene subset (Bone60 gene set). This group includes genes that encode for osteogenic transcription factors and mediators, collagens, collagen-crosslinking enzymes, and non-collagenous extracellular matrix proteins that support mineralization. Most of these bone genes harbor VitC-dependent 5hmC at their loci, while only a subset displays VitC-sensitive H3K27me3 or H3K9me3 demethylation. Consistent with these findings, the majority of bone genes are expressed in a VitC-dependent, 5hmC-dependent fashion, while H3K27me3 or H3K9me3 demethylation appears to play only a minor, gene-selective role in VitC-controlled gene regulation. Both, promoter sites and distal enhancers including super-enhancers, show VitC-dependent dynamics particularly for 5hmC. To explain the primary role of 5hmC and 5hmC-generating enzymes and the more limited function of histone demethylases in orchestrating osteogenic differentiation, we propose that while 5hmC generation acts as an epigenetically active mark designated to facilitate gene expression, demethylation of H3K27me3 or H3K9me3 merely reduces or neutralizes repressive epigenetic marks; and without establishing activating signals such as H3K27ac or H3K9ac, gene expression may still be restricted. Indeed, histone acetyltransferases which would generate such activating epigenetic marks have not been described as VitC-dependent. The latter rationalization may explain why we find 5hmC and 5hmC-generating enzymes to orchestrate a variety of physiological aspects during osteogenic differentiation, while the relevance of histone demethylases is more restricted.

In summary, our study suggests that the skeletal manifestations of scurvy and VitC deficiency represents a micronutrient-dependent epigenetic disease with secondary biochemical effects on collagen maturation. The recognition that VitC epigenetically programs the bone cell fate by both DNA hydroxymethylation and histone demethylation opens opportunities to pharmacologically facilitate bone health and prevent bone degeneration. For example, when considering that VitC supplementation during estrogen replacement therapy in post-menopausal women further increases BMD[32], such strategies could complement the existing standard of care that is currently applied in a broad spectrum of degenerative bone conditions that benefit from bone anabolic strategies.

## Methods
### Mouse models
Effects of Vitamin C (VitC) on bone were studied on homozygous B6.129P2-*Gulo*tm1Mae mice (*Gulo*−/−), which like humans carry a recessive mutant *Gulo* allele that prevents generation of endogenous VitC. These mice were supplemented with Vitamin C (3.3 g/L in drinking water ad libitum, refreshed twice a week) until week 15 (VitC +/−) or until end of the experiment at 20 weeks (VitC+). With proper VitC supplementation in the drinking water, these mice are viable, fertile and indistinguishable from their heterozygous or wild-type littermates[44]. Mice were kept on a standard mouse chow devoid of Vitamin C (LabDiet, #5053). For some experiments, these mice were crossed with C57BL/6-Tg(Dmp1-EGFP_Topaz)1Ikal/J mice[74] expressing an enhanced green fluorescent protein (Topaz) under control of the osteocyte-specific *Dmp1* promoter. Mice containing conditional *Tet1*fl/fl, *Tet2*fl/fl or *Tet1*fl/fl and *Tet2* fl/fl (*Tet1/2*fl/fl) alleles were used to evaluate the role of *Tet1* and *Tet2* in bone tissue. The *Tet1*fl/fl mouse model harbors two loxP sites flanking exon 4. The *Tet2*fl/fl mouse model harbors two loxP sites flanking exon 3 and was purchased from The Jackson Laboratory (*Tet2*tm1.1Iaai; #017573). *Tet1/2*fl/fl mice were generated by breeding both mouse models. *Tet1* and *Tet2* expression was conditionally ablated by mating with mice expressing Cre recombinase from the *Prrx1* enhancer (B6.Cg-Tg) (*Prrx1*-cre)1Cjt/J (JAX #005584); for all experiments only homozygotic mice were used and all mice were on a C57Bl/6 genetic

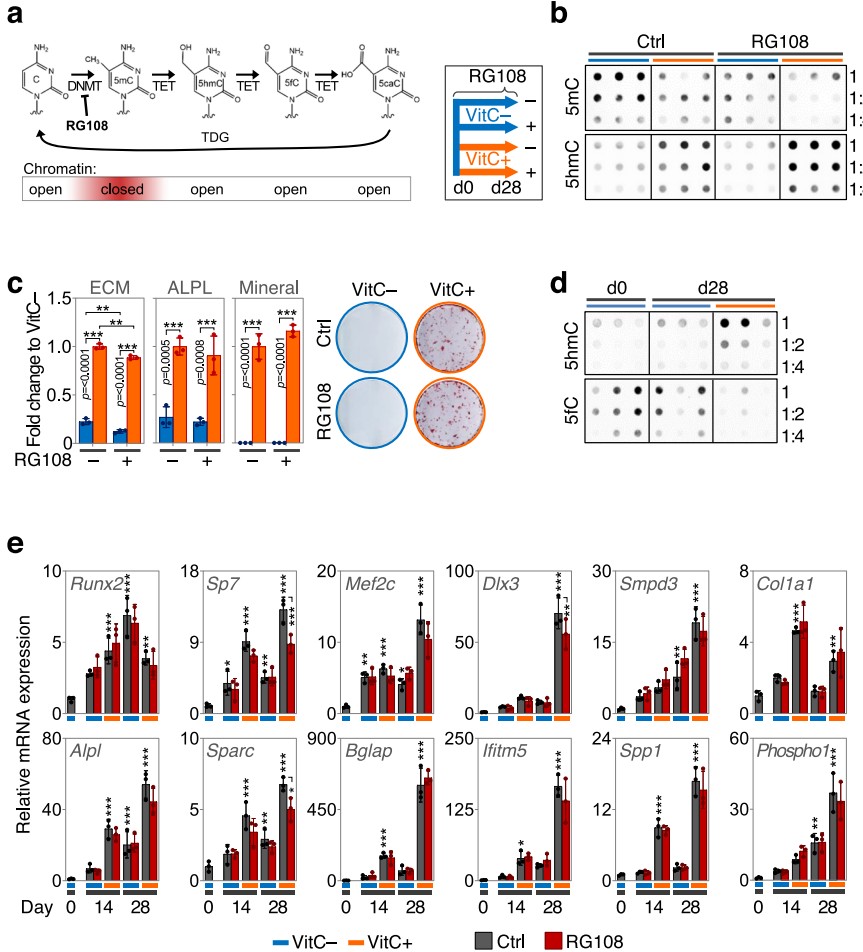

**Fig. 9 | Vitamin C-induced 5hmC is a stable epigenetic mark that is required for osteogenic differentiation. a** Schematic representation of the active DNA demethylation cycle and experimental design for the shown experiments in differentiating osteoblasts. **b** Dot blots at day 14 with and without VitC and/or the DNMT-inhibitor RG108, which decreases global CpG methylation levels (5mC). **c** ECM deposition, alkaline phosphatase activity (ALPL) and ECM mineralization in the presence or absence of VitC and/or RG108. **d** Dot blots for 5hmC and 5fC (formylcytosine) in osteoblasts at indicated time points in the presence or absence of VitC. **e** Osteoblastic gene expression after VitC and/or RG108 administration. Bar graphs represent mean ± SD; *$p < 0.05$; **$p < 0.01$; ***$p < 0.001$. Two-way ANOVA analysis with Sidak's multiple comparison tests (**c** and **e**), groups compared to day 0 Ctrl or as marked in **e**. In **c**, for ECM RG108- VitC- vs RG108 + VitC- **$p = 0.0062$ and RG108-VitC+ vs RG108 + VitC + **$p = 0.0029$. $N = 3$ (**c**, **e**) per group, from biologically independent experiments. Source data as well as exact $p =$ values for all comparisons in (**e**) are provided in the Source Data File.

background. The here presented studies compare wt/wt: *Prrx1*-Cre[+] (Ctrl) versus fl/fl: *Prrx1*-Cre[+] animals and mice are referred in the text as *Tet1* cKO (*Tet1*fl/fl: *Prrx1*-Cre[+]), *Tet2* cKO (*Tet2*fl/fl: *Prrx1*-Cre[+]) and *Tet1/2* cKO (*Tet1/2*fl/fl: *Prrx1*-Cre[+]). The following primers were used for mouse genotyping. *Gulo*[−/−] mice: P2 CGCGCCTTAATTAAGGATCC, P3 GTCGTGACAGAATGTCTTGC and P4 GCATCCCAGTGACTAAGGAT; EGFP detection: forward ATGGTGAGCAAGGGCGAGGAG, reverse TTACTTGTACAGCTCGTCCATG; *Tet1*fl/fl mice: forward 1 (*Tet1* 2ndLoxP F1) TGTTGAGAAAAACGGCACTG, forward 2 (*Tet1* neoGT F1) TCGACTAGAGCTTGCGGAAC, reverse (*Tet1* 2ndLoxP R1) GATAGACCACGTGCCTGGAT; *Tet2*fl/fl mice as recommended by the provider; Cre detection: forward TCCAATTTACTGACCGTACACCAA, reverse CCTGATCCTGGCAATTTCGGCTA. All mice were housed in a selected pathogen-free barrier environment with ad libitum access to food and water, 12-h light and dark cycles, temperature between 20–26 C and humidity at ~70%. All experimental procedures involving laboratory mice were reviewed and approved by the Institutional Animal Care and Use Committee of the Mayo Clinic.

### Microcomputed tomography (μCT) analysis
For all analyzed mouse models, quantitative analyses of the fifth lumbar vertebra (L5) fixed in 4% formaldehyde were performed. For the *Gulo*[−/−] mouse experiments, vertebra from WT ($n = 14$, 7 males and 7 females), VitC + ($n = 11$, 5 males plus 6 females) or VitC ± *Gulo*[−/−] ($n = 9$, 4 males plus 5 females) mice were measured. Similarly, for the conditional Tet1, Tet2, and Tet1/2 mouse models, vertebra from 6 male mice per group were measured. Analyses were performed using a VivaCT 40 scanner (SCANCO Medical AG) with the following parameters: E = 55 kVp, I = 145 μA, and integration time = 300 ms. A voxel size of 10.5 μm using a threshold of 220 units was applied to all scans at high resolution. Using SCANCO Reconstruction software, two-dimensional data from scanned slices were used for a three-dimensional interpolation and calculation of morphometric parameters that define trabecular bone mass and micro-architecture.

### Testing of mechanical proprieties of bone
A three-point bend fixture was mounted on a servohydraulic mechanical testing frame (Model 312, MTS Systems, Eden Prairie, MN) instrumented with a 100-N capacity load cell (Model 3397-25, Leebow Products, Troy, MI). Femurs from 20 week old male WT ($n = 5$), male VitC + ($n = 5$) or male VitC ± ($n = 4$) *Gulo*[−/−] mice were mounted on supports spanning 10 mm. Point loading was applied to the femur midshaft allowing the bones to flex about an axis aligned

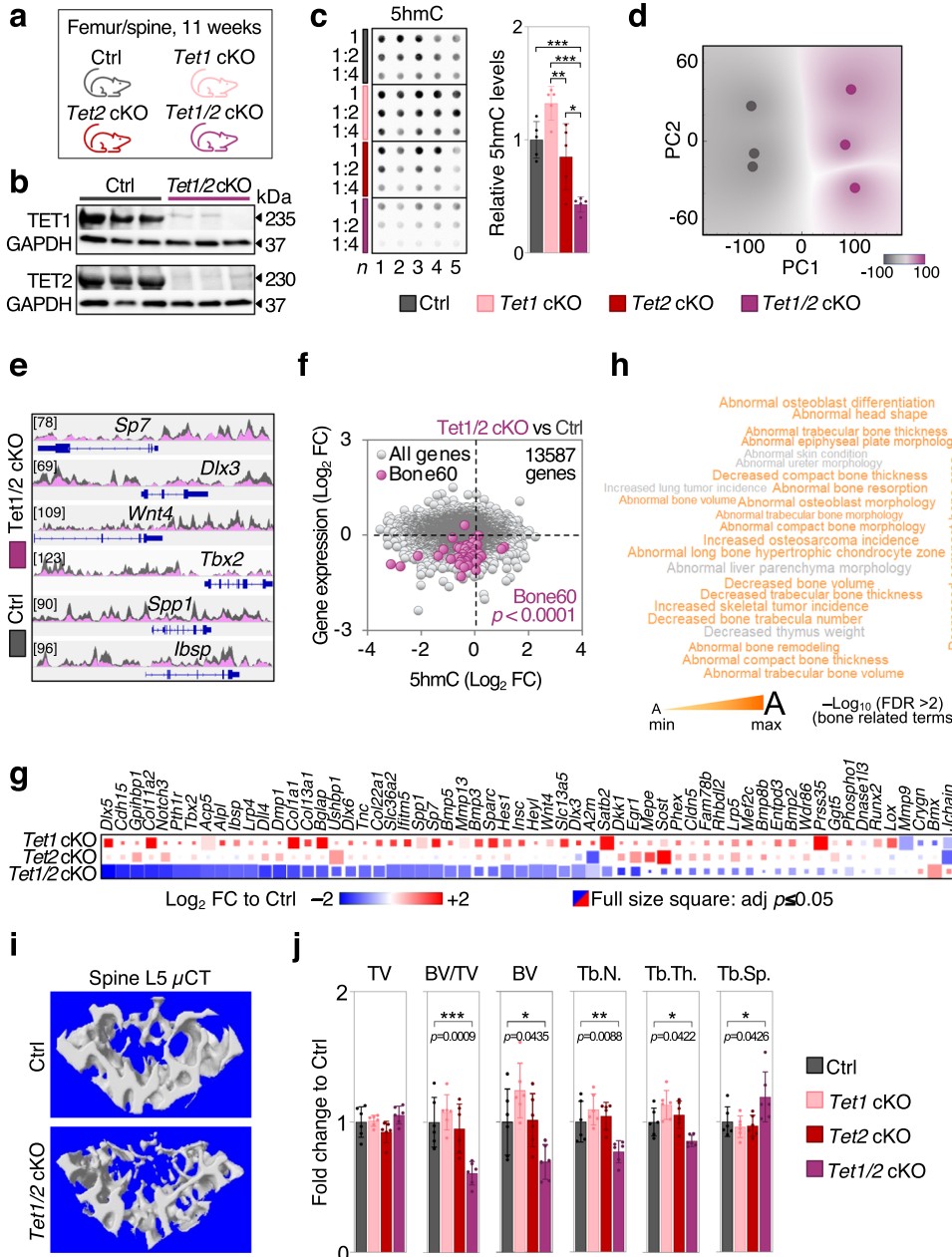

**Fig. 10 | Conditional *Tet1* and *Tet2* deletion impairs DNA hydroxymethylation at bone related loci and causes bone loss in mice. a** Schematic representation of the mouse models used. Bone tissue was collected at 11 weeks; cKO, conditional knockout. **b** *Prrx1* driven TET1 and TET2 knockdown efficiencies in femurs of Tet1/2 double knockout mice. **c** 5hmC dot blot and relative quantification. **d–f** hMeDIP-Seq of femurs from Tet1/2 double knockout mice. **d** Principal component (PC) analysis and 5hmC peaks around selected bone specific genes (**e**). **f** Correlation between 5hmC occupancy (TSS +/−30 kb) and gene expression as measured by RNA-Seq; FC, fold change. **g** Expression of Bone60 genes in conditional *Tet1*, *Tet2* or *Tet1/2* femurs versus Ctrl femurs. **h** Wordcloud representing phenotypes associated with distal intergenic loci with the strongest 5hmC loss in double knockout vs Ctrl femurs as analyzed by GREAT analysis; FDR, false discovery rate. **i** μCT analysis of L5 spine and relative quantification; TV total volume, BV bone volume, Tb.N. trabecular number, Tb.Th. trabecular thickness, Tb.Sp. trabecular separation (**j**). Bar graphs represent mean ± SD; *$p < 0.05$; **$p < 0.01$; ***$p < 0.001$. One-way ANOVA analysis with Tukey's (**c**) and Dunnett's (**j**) multiple comparison tests. FDR-adjusted two-tailed, unpaired *t* tests in **g**, two-sided Fishers exact test (**f**). In **c**, Ctrl vs *Tet1/2* cKO ***$p = 0.0009$, *Tet1* cKO vs *Tet1/2* cKO ***$p = < 0.0001$, *Tet1* cKO vs *Tet2* cKO **$p = 0.005$, *Tet2* cKO vs *Tet1/2* cKO *$p = 0.0118$. $N = 5$ (**c**) & $n = 6$ per group in **j** from biologically independent animals. Source data are provided as a Source Data File.

with the medial-lateral line. Specimens were kept moist by irrigating them with phosphate buffered saline solution. Loading was applied under displacement control at a rate of 20 mm/min until fracture. Force and displacement data was sampled at 200 Hz. The peak load was quantified as was the peak displacement. The stiffness was calculated from the slope of the linear region of the force displacement curve. With the exception of one removed specimen for the VitC +/− group, all failures occurred at the femur midshaft below the point of load application.

## Assessment of RNA expression

Total RNA from flushed and minced femoral bones or cell cultures was isolated using the Direct-Zol RNA Mini Prep Kit (Zymo Research) following the supplier's instructions. For quantitative real-time polymerase chain reaction (rt-qPCR), cDNA was synthesized from ~0.5 μg RNA using the SuperScript III first strand synthesis system (Invitrogen) as described by the supplier and subjected to rt-qPCR amplification using the QuantiTect SYBR-Green PCR Kit (Qiagen) and a CFX384 real time system machine (Bio-Rad) using the following conditions: 10 min

95 °C followed by 45 cycles of 30 sec at 95 °C, 30 sec at 60 °C and 30 s at 72 °C. Primers for the analyzed genes are listed in Table S1. All rt-qPCR assays were performed in triplicate and expression was evaluated using the Biorad CFX Manager 3.0 software applying the comparative quantification method[75]. For high through output next generation RNA sequencing (RNA-seq) RNA integrity was analyzed with a 2100 Bioanalyzer with an RNA 6000 kit (Agilent Technologies). RNA-Seq libraries were prepared using the Illumina TruSeq v2 library preparation kit and sequenced with paired-end 50 bp reads on an Illumina HiSeq 4000 in biological triplicates for bone tissues (femurs) and duplicates for cell-lines.

### Immunofluorescence of calvarial bone sections

Calvarial bones from 20 week old male VitC+ or VitC ± *Gulo*$^{-/-}$ mice crossed with C57BL/6-Tg(*Dmp1*-EGFP$_{Topaz}$)$^{1Ikal/J}$ were fixed in 4% paraformaldehyde at 4 °C for 48 h. Next, samples were decalcified with 14% EDTA (pH 7.2) at 4 °C until complete decalcification and embedded for cryo-sectioning. Coronal plane cuts of about 30 μM were washed 3 times in PBS for 5 min before permeabilization with RPM in PBS for 2 h at room temperature. Subsequently samples were incubated in 2 N HCl for 1 h and then quickly washed once with water and twice with 100 mM Tris-HCl pH 8.0 for 10 min before blocking in 1% donkey serum (Abcam, ab7475) at 4 °C overnight. Next day, calvarial sections were incubated with a 5hmC antibody (Active Motif, #39769, 1:5,000) or normal rabbit IgG (Santa Cruz, #sc-2027, 1:100) in 1% donkey serum in PBS overnight. Sections were then washed 5 times for 5 min with PBS and incubated overnight with a goat anti-rabbit Alexa-647 conjugated secondary antibody (Thermo Fisher, #A-21245) diluted 1:400 in 1% donkey serum in PBS. Subsequently samples were again washed 5 times with PBS for 5 min before incubation with Hoechst 33258 dye for nuclear staining for 1 h. Before mounting, sections were incubated in increasing concentrations of 2,2′-thiodiethanol (TDE, Sigma-Aldrich) and finally mounted in 100% TDE. All here mentioned incubation steps were performed in the dark. Laser scanning microscopy was performed with a 40x objective on a Zeiss LSM 780 microscope in conjunction with Zen blue edition software version 2.3 or Zen black edition software version 14.0 visualizing nuclei in blue, EGFP expression in green, 5hmC signal in red and bone shape by transmission. Fluorescence images are shown as maximum projections of Z-stacks of approximately 15 μM in depth using 0.4 μM imaging steps.

### Immunofluorescence of cell cultures

For 5hmC detection, cells were fixed with 4% paraformaldehyde for 20 min, for BGLAP detection, cells were fixed in ice-cold methanol at −20 °C for 10 min. Cells were then permeabilized with 0.5% Triton in 4% paraformaldehyde for 5 min at room temperature. For 5hmC detection, chromatin was denatured with 2 N HCl for 20 min and then washed twice with 100 mM Tris-HCl pH 8.0 for 5 min. Blocking was performed in 1% donkey serum (Abcam, ab7475) for 1 h. Cells were then incubated with the selected antibodies (5hmC, Active Motif, #39769, 1:5,000; BGLAP, Abcam, ab13421, 10 μg/ml) or normal rabbit IgG (Santa Cruz, #sc-2027, 1:100) in 1% donkey serum in PBS overnight and then with an appropriate secondary antibody (goat anti-rabbit Alexa-647 or goat anti-mouse Alexa-488, Thermo Fisher; A-21245 and A-11001, respectively) diluted 1:400 in 1% donkey serum in PBS for 1 h. Adipogenic lipid droplet formation was visualized with LipidSpot (Biotium) as per manufacturer's recommendations. Image acquisition and analysis was performed with a Zeiss LSM 780 microscope or a Zeiss Axio Vert.A1 microscope in conjunction with Zen blue edition software version 2.3 or Zen black edition software version 14.0 as well as ImageJ software (National Institute of Health, Bethesda, MD).

### Dot blot analysis

DNA was extracted using a Qiagen DNA extraction kit following the manufacturer's instructions. 1 μg of heat denatured genomic DNA (gDNA) diluted in 0.1 M NaOH/1 M ammonium acetate was spotted in 1:2 dilution steps onto a positively charged nylon membrane (Biorad) and rinsed in 2× SSC buffer followed by complete air dry and 30 min baking at 80 °C. After blocking with 5% non-fat dry milk in PBST, the membrane was immunoblotted using 5mC (Diagenode, C15200081, 1:300), 5hmC (Active Motif, 39791, 0.2 μg/ml) or 5fC (Millipore, MABE1092 FC-5, 0.5 μg/ml) antibodies diluted in 5% non-fat dry milk in PBST overnight. Next day, the membrane was washed three times with PBS for 5 min and incubated with appropriate HRP-conjugated secondary antibody (Cell signaling, 7074 and 7076, 1:5000) in 5% non-fat dry milk in PBST overnight for 1 h, washed three times with PBS and finally developed with enhanced chemiluminescence reagents (Thermo Scientific, 34095), exposed with a ChemiDoc Touch Imaging System and quantified with the corresponding software Biorad Image Lab software, version 5.2.1.

### 5hmC MeDIP-Seq

gDNA was extracted from decalcified femurs from VitC+ and VitC +/− *Gulo*$^{-/-}$ mice (*n* = 3 per group), decalcified femurs from Ctrl or *Prrx1-Tet1/2* double KO mice (*n* = 3 per group) or from cultured VitC+ and VitC- MC3T3-E1 cells (*n* = 2 per group) using a Qiagen DNA extraction kit following manufacturer's instructions. gDNA was then sonicated with a Bioruptor Pico (Diagenode, Seraing, Belgium) at a concentration of 10 ng/μl for 15 cycles of 30 sec on and 30 sec off to an average fragment size of 300 bp. After sonication, gDNA was concentrated using Amicon Ultra centrifugal filter units (Millipore). 4 μg of gDNA was heat denatured and pre-incubated with 4 μg of anti-5hmC antibody (Active Motif, 39791) or 2.5 μg of control rabbit IgG (Santa Cruz, sc-2027) for 3 h before adding protein G Dynabeads (Thermo Fisher) and incubation on a rotator at 4 °C overnight. Beads-antibody-DNA complexes were captured and washed on a magnetic rack and then 5hmC enriched DNA fragments were eluted from the beads, purified with the ssDNA/RNA Clean & Concentrator Kit (Zymo Research) and quantified using Qubit ssDNA High Sensitivity assay (Thermo Scientific). The libraries were prepared by the ACCEL-NGS® 1 S Plus DNA library kit (Swift Bioscience, MI, Ann Arbor) following manufacturer's instructions and sequenced to 51 base pairs from both ends using the Illumina HiSeq 4000 instrument at the Mayo Clinic Medical Genome Facility Sequencing Core.

### Cell cultures

All cell cultures received media changes at three-day intervals and were maintained at 37 °C in a 5% CO$_2$ humidified atmosphere. Primary cultures of mouse bone marrow stromal cells (mBMSCs) were isolated from 20 week old VitC +/− Dmp1-EGFPTopaz/*Gulo*$^{-/-}$ mice using established protocols[76]. Briefly, bone marrow was flushed from femurs and tibias and cells were collected and plated and expanded in Vitamin C free αMEM (Gibco) cell culture media supplemented with 10% FBS and 1% penicillin-streptomycin (Gibco). Primary cultures of human bone marrow stroma cells (hBMSCs) from multiple donors were purchased from Lonza (PT-2501) and expanded in Vitamin C free αMEM (Gibco) cell culture media supplemented with 10% FBS and 1% penicillin-streptomycin (Gibco), all experiments were performed with cells at passage 3. For osteogenic differentiation, cells were seeded at 20,000 cells/cm$^2$ and cultured for the indicated times with or without Vitamin C (50 μg/ml) and with 4 mM β-glycerophosphate (βgp). Inhibition of collagenous extra cellular matrix (ECM) deposition was achieved by treating the cells with 12.5 μM of 1,4-DPCA as this concentration ECM deposition is similar to Vitamin C free conditions.

For adipogenic differentiation, after seeding at 20,000 cells/cm$^2$, BMSCs were cultured with or without increasing concentrations of Vitamin C in high-glucose DMEM supplemented with 10% FBS and 1% pen-strep, 0.86 μM human insulin, 0.25 μM dexamethasone, and 5 mM isobutylmethylxanthine (IBMX; all Sigma) for the first 4 days.

Thereafter, cells were cultured without dexamethasone or IBMX in the media.

Calvarial mouse pre-osteoblastic MC3T3-E1 cells (ATCC CRL-2593, subclone 4) were propagated in Vitamin C free αMEM supplemented with 10% FBS and 1% penicillin-streptomycin (Gibco). For experiments, cells were seeded at a density of 10,000 cells/cm$^2$ and cultured for the indicated times with or without Vitamin C (50 μg/ml) and 4 mM βGP. For inhibition of DNMTs, cells were treated with 100 μM RG108 (Tocris) and for inhibition of αKGDDs, MC3T3-E1 cells as well as primary mouse BMSCs were treated with 20 mM (**2** L)-**2**-Hydroxyglutaric Acid Octyl Ester Sodium Salt (2-L-HG, Toronto Research Chemicals). For antioxidant experiments, mBMSCs were treated with 3 mM N-acetyl-cysteine (NAC; Santa Cruz biotechnologies).

The murine osteoblast to osteocyte-like MLO-A5 cell line (a generous gifted from Dr. Lynda Bonewald, Indiana University School of Medicine) was maintained on collagen coated plates (0.15 mg/ml rat tail type I collagen) using Vitamin C free αMEM supplemented with 5% FBS and 5% CS and penicillin-streptomycin (Gibco). For differentiation purposes, cells were seeded on collagen coated plates at 10,000 cells/cm$^2$ in αMEM supplemented with 10% FBS and penicillin-streptomycin (Gibco), 50 μg/ml Vitamin C and 4 mM βgp and cultured for the indicated periods.

## Stable gene knock down by shRNA and viral particle generation

HEK293 packaging cells (ATCC, CRL-1573) were co-transfected with MISSION TRC2-pLKO-puroR vectors (Sigma; empty vector control (EVCtrl)), scramble shRNA (NegCtrl) or shRNAs against *Tet1*, anti *Tet2*, anti *Tet3*, *Kdm4a*, *Kdm4b*, *Kdm4c*, *Kdm6a*, *Kdm6b* and *Jhdm1d*, packaging plasmid psPAX2 (Addgene plasmid #12260) and envelope plasmid pMD2.G (Addgene plasmid #12259) with Fugene transfection reagent (Promega). Lentiviral supernatants were collected at 48, 72, and 96 h. mBMSCs, MC3T3-E1 and MLO-A5 cells were transduced three times by spin infection with lentiviruses with a time gap of 24 hr. Cells expressing puroR were selected by treatment with puromycin (Santa Cruz) after the third infection. Efficiencies of shRNA knockdowns were tested beforehand by rt-qPCR and the two most efficient shRNAs were selected for experiments.

## Endogenous gene overexpression

Endogenous TET2, KDM4A, KDM4B, KDM4C, KDM6A, KDM6B and JHDM1D overexpression was induced using a gRNA directed, tetracycline inducible dCas9-VP64 system (Addgene plasmids #50916, #50920) in MC3T3-E1 cells. For each gene, a screen of 5 or more promoter binding gRNAs was performed and the two gRNAs showing the highest induction activity were chosen for further experiments, as negative control the empty backbone plasmid was used (#50920). Viral particles generation and cell transduction were performed as described above, gRNA sequences are listed in Table S1.

## Alkaline phosphatase (ALPL) activity

ALPL activity was determined in relation to total DNA present in samples as a surrogate for cell number. Cells were washed with PBS and frozen in 1 mM Tris–HCl buffer (pH 8.0) containing 0.1 mM EDTA. During thawing, DNA content was determined using 1 μg/ml Hoechst 33258 dye (Polysciences, Warrington, PA). After incubation of 15 min at room temperature, the fluorescence intensity was measured (excitation 360, emission 465 nm). The amount of DNA was estimated using a standard curve prepared from calf thymus DNA (Roche). ALPL activity was quantified with p-nitrophenylphosphate (2.5 mg/ml in 0.1 M diethanolamine buffer [pH 10.5], 150 mM NaCl, 2 mM MgCl$_2$) by incubation of the cells for 15 min at room temperature and measurements were referenced to a standard curve prepared from calf intestinal ALP (Roche). Absorption was measured in a microplate reader at 405/490 nm (Tecan) and the instrument's software Magellan version 7.2. ALPL activity was expressed as arbitrary units per milligram DNA and referenced to fold change to control.

## Extra cellular matrix quantification and visualization by Picro Sirius Red staining

Cultured cells were dislodged from the extracellular matrix (ECM) using 0.5% sodium deoxycholate in PBS for 20 min at 4 °C. The remaining ECM was thoroughly washed with PBS and ECM collagen staining was performed by Picro Sirius Red staining as shown before[77]. Fluorescence quantification of Picro Sirius Red stained ECM collagen matrix was performed on a multi-plate reader (Tecan) at 535 nm/ 633 nm using the instruments software Magellan version 7.2. Visualization of ECM collagen structure was performed by laser scanning microscopy on a Zeiss LSM 780 instrument (561 nm/628 nm) in conjunction with Zen blue edition software version 2.3 or Zen black edition software version 14.0.

## Collagen crosslinking

After 7 days in culture MC3T3-E1 cells were washed with ice cold PBS and fixed in 75% ethanol. Cell pellets were used for spectroscopic analysis by FTIR Imaging for the determination of pyr/divalent collagen cross-link ratio. The pellets were transferred onto barium fluoride windows, where they were air-dried. Following this, spectra were obtained in transmission with a Bruker (Germany) Equinox 55 spectrometer coupled to a Bruker Hyperion 3000 FTIR microscope equipped with a motorized stage (±1 μm) and a 15× objective. The spectra were baseline-corrected in the amide I and II spectral area (~1500–1700 cm$^{-1}$); water vapor was subtracted and then subjected to second derivative spectroscopy and curve fitting routines as described elsewhere. The collagen cross-link ratio was determined as previously described[78].

## Assessment of ECM mineralization by Alizarin Red staining

Cells subjected to Vitamin C dependent osteogenic differentiation were fixed for 20 min in 4% paraformaldehyde and mineralization was measured by Alizarin Red S stain (Sigma-Aldrich). Mineralization deposits were quantified by digital image analysis using ImageJ.

## Assessment of fat droplet formation by Oil Red O staining

After cell culture, cells were rinsed twice with PBS and then fixed with 4% paraformaldehyde in PBS for 20 min at room temperature. Thereafter fixed cells were rinsed three times with distilled water and incubated with 2% Oil Red O (Sigma) in 60% isopropanol solution for 30 min. Subsequently, cells were rinsed five times with distilled water and dried for 10 min at 32 °C. Finally, Oil Red O stain was extracted with isopropanol and absorbencies were measured at 490 nm with a multi-well plate reader (Tecan) together with the instrument's software Magellan version 7.2.

## Assessment of Colony forming cell units (CFU)

BMSCs were isolated from 20-week-old VitC supplemented (VitC+) or from week 15 to week 20 VitC deprived (VitC ±) *Gulo*$^{-/-}$ mice as described above and seeded at a density of 150,000 cells/cm$^2$ into 6 multiwell-plates. Colony formation assay was performed by using the MesenCult™ Expansion Kit following manufacturer's recommendations. After 14 days of culture, cells were fixed in 4% PFA and stained with a solution of 0.5% crystal violet and 25% methanol in 1x PBS for 2 h. To remove staining excess wells were rinsed with tap water for at least 5 times. Images were digitalized and colonies were analyzed using the ImageJ software.

## Western blot analysis

Cell cultures were washed three times with ice cold PBS and lysed in SDS buffer (3% SDS v/v in 0.25 M Tris pH 6.8). Total protein amounts were quantified by BCA protein assay (Sigma-Aldrich) and protein

concentrations were equalized between the samples subsequently. For protein separation on SDS-PAGE 8% or 11% Bis-Tris gels, 8 μg protein/sample were mixed with Loemmli buffer, boiled at 95 °C for 5 min and resolved on the gel. Proteins were transferred to a nitrocellulose membrane and subsequently Ponceau S stain was performed to monitor equal protein loading between the samples. Membranes were then blocked in 5% skimmed milk in TBST for 1 h and further incubated with primary antibody overnight (H3K9me3, Active Motif, 39062, 1:3000; H3K27me3, Cell Signaling, 9733, 1:1000; TET1, Genetex, GTX124207, N3C1, 1:1000; TET2, Abcam, ab124297, 1:250; KDM4C, Novus Biologicals, NBP1-49600, 1:10000; KDM6A, Novus biologicals, NBP1-80628, 0.1 μg/ml; GAPDH, Cell signaling Technology, 5174, 1:1000; total histone 3, Millipore-Sigma, 06-755, 1:5000). Next day, membranes were washed 3 times with TBST and incubated with anti-rabbit horseradish peroxidase-conjugated secondary antibody (Cell signaling, 7074, 1:5000). Proteins were visualized using enhanced chemiluminescence reagents (Thermo Scientific, 34095), exposed with a ChemiDoc Touch Imaging System (Biorad). Data acquisition and analysis was performed with the Biorad Image Lab software, version 5.2.1. Uncropped Western blot scans for the Main Figures are available as a Source Data File and uncropped Western Blot scans for the Supplementary Figures are provided at the end of the Supplementary Information file.

### Dmp1 promoter activity assessment in differentiating BMSCs

Cells from 20 week old VitC ± Dmp1-EGFPTopaz/Gulo$^{-/-}$ mice were isolated from long bones and seeded at a density of 20,000 cells/cm$^2$ as described above. EGFPTopaz signal in living cells was measured and quantified using a multiplate reader (Tecan) at 37 °C and 5% CO$_2$ every five days from day 35 to day 50. Fluorescence signal was subtracted to media only control wells. Data analysis was performed with the Tecan Magellan software version 7.2

### RNA-seq data analysis

Raw reads from RNA-seq experiments were assessed for run quality using fastqc version 0.72, followed by alignment to reference genome (mm10 for mouse and hg19 for human) using RNA STAR Galaxy version 2.6.0b-1 using default settings. Differential gene expression analysis was performed using featureCounts Galaxy version 1.6.3 and DESeq2 Galaxy version 2.11.40.2. Fragments per kilobase of exon per million mapped fragments (FPKM) was performed by cufflinks v2.2.0. Hierarchical clustering was executed with Morpheus (Broad Institute, https://software.broadinstitute.org/morpheus) using Euclidean distance as metric. For the definition of Bone60 and Fat76 gene-sets as well as tor the comparison of the transcriptomes between mESCs and mBMSCs, raw-files from publically available RNA-Seq datasets from different human and/or control wildtype mouse tissues or cells (table S2) were downloaded from the NCBI-SRA database, assessed for read quality with fastqc version 0.72 and analyzed by organism as described above. To further normalize for possible batch effects between the samples, RUVSeq (Galaxy Version 1.26.0) was applied in conjunction with DESeq2. Bone or fat tissue-related gene-sets as defined by DESeq2 and including already well-known bone and fat related genes, were then compared between human and mouse and only genes found in both organisms were selected for the final gene-sets. The generated gene-sets were then used for gene-set enrichment analysis (GSEA) and other studies. GSEA analysis was performed with GSEA version 4.0.1 (Broad Institute) using RNA-Seq datasets from femurs from 20 week old VitC+ and VitC ± Gulo$^{-/-}$ mice (n = 3 per group).

### 5hmC MeDIP-seq data analysis

Raw reads from 5hmC MeDIP-seq (hMeDIP-seq) experiments were assessed for quality using fastqc version 0.72, followed by mapping to reference genome (mm10) using BWA Galaxy version 0.7.15.1 with

standard settings and pooled (n = 3 per group) peak calling was performed with MACS2 callpeak version 2.1.0.20140616.0 with standard settings including a minimum FDR cutoff of 0.05. Peak size and location was visualized with IGV version 2.3.92.MAYO by overlaying the tracks and using group auto-scale for comparable visualization of both groups. Peak number around transcriptional start sites (TSS, ±30 kbp) was calculated in 2500 bp bins and 5hmC intensity around TSS (±3 kbp) and over to 4 kbp normalized gene bodies as well as genomic 5hmC distribution was assessed with CEAS software (Version1.0.0, Cistrome, Liu Lab). To assess significantly different peak binding between two experimental groups, peak files (MACS2) from single hMeDIP-seq runs were analyzed with DiffBind Galaxy version 2.6.6.4 or 2.10.0 using standard settings. Super-enhancer (SE) like analysis on 5hmC peak distribution was performed by using the package ROSE and the software NaviSE[79]. Gene-ontology analysis for gene loci near 5hmC positive cis-regulatory distal intergenic regions was performed using GREAT version 3.0.0 (Bejerano Lab, Stanford)[80]. The top 10% (9502 significant peaks in VitC +/− versus VitC+ Gulo$^{-/-}$ mice) of the most significantly (FDR, DiffBind) decreased 5hmC peaks were used for the analysis.

### ChIP-seq assays and ChIP-Seq data analysis

Raw-files from publically available H3K27ac ChIP-Seq datasets from wildtype mice BMSCs differentiated into osteoblasts for 21 days (Table S2, n = 2) were downloaded from NCBI-SRA database. For H3K9me3 and H3K27me3 ChIP-seq assays and analysis, MC3T3-E1 cells were seeded at 20,000 cells/cm$^2$ in 10 cm dishes in duplicates and treated with or without VitC for 7 days. Thereafter, cells were fixed with 1% paraformaldehyde (PFA) for 10 min and then subjected to ChIP-Seq using a rabbit anti-H3K9me3 antibody (Diagenode, C15410056) or a rabbit anti-H3K27me3 antibody (Cell Signaling Technology, 9733BC, C36B11) in the Mayo Clinic Epigenomics Development Laboratory as described before[81]. The ChIP-seq libraries were sequenced to 51 bp from both ends on an Illumina HiSeq 4000 instrument in the Mayo Clinic Medical Genomics Core Facility. Raw ChIP-Seq files were assessed for read quality with fastqc version 0.72 and mapped to the mouse genome (mm10) as described above. Peak calling was performed with MACS2 callpeak version 2.1.0.20140616.0 using broad peak settings and including a minimum FDR cutoff of 0.05. To assess significantly different peak binding between two experimental groups, peak files (MACS2) from single ChIP-seq runs were analyzed with DiffBind Galaxy version 2.6.6.4 or 2.10.0. Peak size and location was visualized with IGV version 2.3.92.MAYO using group auto-scale for comparable visualization of all shown groups. Peak number around transcriptional start sites (TSS, ± 30 kbp) was then calculated in 2500 bp bins and peak intensity around TSS (±3 kbp) and over to 4 kbp normalized gene bodies as well as genomic peak distribution was assessed with CEAS software (Version1.0.0, Cistrome, Liu Lab). Gene-ontology analysis for gene loci near H3K9me3 and H3K27me3 positive cis-regulatory distal intergenic regions was performed using GREAT version 3.0.0 (Bejerano Lab, Stanford)[60]. The most significantly (FDR, DiffBind) increased H3K9me3 or H3K27me3 peaks in after VitC treatment were used for the analysis. For H3K27ac sequencing files, super enhancer analysis was performed by using the package ROSE as described above. Overlapping SE between 5hmC SE from VitC treated MC3T3-E1 cells (D28) and H3K27ac SE from BMSCs were filtered with Bedtools (version 2.29) using a minimum overlap of 50%.

### Statistics and reproducibility

The vast majority of statistical data analysis and figure generation was performed with GraphPad Prism software, version 8. Statistical analysis for RNA-Seq datasets was performed by DeSeq2 Galaxy version 2.11.40.2. All experiments were performed at a minimum of independent biological and/or experimental triplicates as indicated in the figures and figure legends. Where representative pictures are shown,

pictures showing similar results were taken from at least three independent experiments.

**Reporting summary**

Further information on research design is available in the Nature Research Reporting Summary linked to this article.

## Data availability

The raw and processed RNA-Seq, MeDIP-Seq and ChIP-Seq data-sets generated in this study have been deposited in the GEO database under accession code GSE138854. All other data generated in this study are provided in the Supplementary Information files and the Source Data file. Source data are provided with this paper.

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

## Acknowledgements

We thank the members of our laboratories, including Markus Schreiner, Akane Shu, Nolan Vale, Miriam Galvan Tenorio and Chris Paradise, our institutional colleagues Jennifer Westendorf, Matthew Abdel, Dana Begun, Siva Arumugam Saravanaperumal, Huihuang Yan, and Krutika Gaonkar, as well as our extramural collaborators Paul Roschger, Klaus Klaushofer, Franz Varga and Gary Stein for stimulating discussions and support. Funding: These studies were supported by NIH grant R01 AR049069 (to A.J.v.W.), as well as a fellowship award from the Mayo

Clinic Center of Regenerative Medicine (to R.T.) and the Patrick J. Kelly Fellowship award (to R.T.). NIH grants RO1 DK58185, RO1 DK126827 and RO1 DK131455 supported T.O.

## Author contributions

R.T. conceived the project. R.T., T.O., and A.J.v.W. conceptualized the study and R.T. designed experiments. R.T. and F.K. performed the majority of data acquisition and analysis with help of I.S., S.S.D., J.M.D., X.Z., O.P., A.D., S.S.J., D.R.D., E.P.P., and B.M.M.. J.Z., J.H.L., and T.O. conducted epigenomics studies. R.M., I.K., and Y.H.J. generated mouse models. R.T., I.S., T.O., and A.J.v.W. conceptualized and wrote the manuscript, while all other authors edited the manuscript. R.T., T.O., and A.J.v.W. supervised various aspects of the work. R.T. and A.J.v.W. acquired funding for these studies.

## Competing interests

The authors declare no competing interests. This research has been reviewed by the Mayo Clinic Conflict of Interest Review Board and was conducted in compliance with Mayo Clinic conflict of interest policies.

## Additional information

[1]Department of Orthopedic Surgery, Mayo Clinic, Rochester, MN, USA. [2]Department of Biochemistry & Molecular Biology, Mayo Clinic, Rochester, MN, USA. [3]Center for Regenerative Medicine, Mayo Clinic, Rochester, MN, USA. [4]Departments of Pediatric and Adolescent Medicine, Mayo Clinic, Rochester, MN, USA. [5]Epigenomics Program, Center for Individualized Medicine, Mayo Clinic, Rochester, MN, USA. [6]Department of Internal Medicine, Virginia Commonwealth University, Richmond, VA, USA. [7]Department of Reconstructive Sciences, UConn Health, Farmington, CT, USA. [8]Department of Genetics, Neuroscience, and Pediatrics, Yale University School of Medicine, New Haven, CT, USA. [9]Department of Clinical Genomics, Mayo Clinic, Rochester, MN, USA. [10]Ludwig Boltzmann Institute of Osteology at Hanusch Hospital of OEGK and AUVA Trauma Centre Meidling, 1st Med. Dept. Hanusch Hospital, Vienna, Austria. [11]Department of Physiology and Biomedical Engineering and Division of Gastroenterology and Hepatology, Department of Medicine, Mayo Clinic, Rochester, MN, USA. [12]Department of Biochemistry, University of Vermont, Burlington, VT, USA. ✉e-mail: Thaler.Roman@mayo.edu; andre.vanwijnen@uvm.edu

