## [Peer Review File · Nature Communications]

Vitamin C epigenetically controls osteogenesis and bone mineralizationReviewers' comments:

Reviewer #1 (Remarks to the Author):

The manuscript "Vitamin C epigenetically controls osteogenesis and bone mineralization" is interesting, the aim is clear, and the methods seem appropriate. Previously similar studies have been published in different systems, but this study has some novel findings related to bone biology. The best part of this study is that the authors used an appropriate mice model (gulono-lactone oxidase, GULO knockout) to study the role of vitamin C (as mice make vitamin C) in bone biology. The findings are novel in the field of bone biology.

The main drawback of this study is that authors performed in vitro studies on mice BMSCs, which lack translational importance. The author should perform one or two key experiments on human BMSCs (which are commercially available) to get some valuable translational data. The discussion of the manuscript needs to be elaborated. It does not tell the entire novel findings generated by authors. Overall, I am happy that reputed scientist Dr. Wijnen investigating vitamin C role in bone biology. There are few labs investigating the important role of vitamin C in musculoskeletal pathobiology.

Reviewer #2 (Remarks to the Author):

Comments to the Authors

The present manuscript entitled "Vitamin C epigenetically controls osteogenesis and bone mineralization" investigates the DNA and histone methylation basis of Vitamin C for bone integrity and osteoblastic differentiation. Although that Vitamin C regulation on osteogenesis and bone formation have been reported and that Vitamin C is known to act as an epigenetic regulator, this manuscript is the first to decipher detailed epigenetic mechanisms underlying effects of Vitamin C on osteogenesis. The major drawback is lack of enough in vivo experimental evidence to explore epigenetic targets.

Following is detailed comments:

Major concerns

1. The title proposes "epigenetically controls bone mineralization", but lack of in vivo functional experiments on epigenetic targets. The authors should at least perform in vivo loss-of-function and gain-of-function analysis on the proposed target, Tet2.
2. In this manuscript cell lines were used in several experiments instead of primary cells. It is risky to draw conclusions from cell lines particularly when the authors try to answer mechanistic questions related to in vivo phenomena. Primary bone marrow mesenchymal stem cells should be used throughout the manuscript for in vitro studies.
3. Stem cell function should be examined using appropriate methods, such as multipotent differentiation and self-renewal assays.
4. It is necessary to verify epigenetic targets in primary mesenchymal stem cells.
5. The authors should strengthen the significance of Vitamin C epigenetically promoting bone formation regarding clinical relevance. For example, in the case of bone defects, can Vitamin C regulate epigenetic mechanisms to improve bone healing? Or can Vitamin C be used to epigenetically promote the stem cell function and increase the therapeutic effects?
6. The results show that Vitamin C promotes osteogenic differentiation. However, it might actually be attributed to a general effect of Vitamin C recovering stemness of cells, as previously reported the pluripotency reprogramming function of Vitamin C. The authors need to check if broader functions of stem cells are affected, such as proliferation, senescence, cell cycle progression and expression of stemness genes.
7. The authors have proved that effects on collagen formation are dispensable for Vitamin C promoting osteogenesis. However, Vitamin C has also been recognized as a potent antioxidant. Whether the antioxidant effects actually underlie the effects of Vitamin C? Controls such as NAC should be used in in vivo experiments and on cultured stem cells. It would be better if the authors perform supplemental

experiments on loss-of-function experiments of antioxidant mechanisms.

8. It is interesting that Vitamin C is dispensable for adipogenesis, as osteogenesis and adipogenesis is often reversely correlated. More discussions are needed to explain the sole requirement of Vitamin C for osteogenesis and why the epigenetic changes do not affect adipogenesis.

9. Although histone methylation is tightly regulated by Vitamin C according to this study, the mechanisms remain unclear. Only Tet2 is revealed responsible for the DNA methylation changes. It is recommended to further dissect mechanisms of Vitamin C regulating histone methylation together with functional experiments of the targets.

10. The introduction and discussion sections lack comprehensive review of previous studies, particularly the application of Vitamin C in stem cell and bone contexts, its role as an epigenetic regulator and the understanding of Tets in regulating mesenchymal stem cell function and bone homeostasis. The study should be based on clear and sound references of the current advance.

Minor concerns

1. There is no functional experiment data about 5hmC, so page5/line104 "by the activating 5hmC" should be changed to "with activating 5hmC".

2. More than half of the references are published at over five years ago and the knowledge of the latest advances in the field is lacking.

Reviewer #3 (Remarks to the Author):

NCOMMS-20-39677-T, Thaler and co-authors, Vitamin C epigenetically controls osteogenesis and bone mineralization

In this study, by using vitamin C-deficient Gulo knockout mouse model, the authors showed that Vitamin C deficiency induces transcriptional changes in bone tissues, with vitamin C-deficient mice displaying a phenotype of severe defects in bone structure. Vitamin C controls the epigenetic program in bone tissue and during osteogenic differentiation by downregulating the levels of repressive histone marks H3K9me3 and H3K27me3, and promoting Tet-mediated 5hmC deposition at bone-specific genes. Vitamin C is required for bone marrow-derived mesenchymal stem cells (BMSCs) differentiation into osteoblast lineage, but not adipocyte lineage. Further, the authors showed that knockdown of TET2 and certain members of KDM/JMJD family demethylases affects Vitamin C-mediated osteogenic differentiation, while overexpression of TET2 promotes osteoblast differentiation.

Overall, the authors presented an interesting study about the epigenetic function of Vitamin C during osteogenic differentiation. The following are some comments:

Major points:

1. The authors showed by western blot that vitamin C deficiency promotes the accumulation of the repressive histone marks H3K9me3 and H3K27me3, however, this only provides data of the overall levels of these histone marks in the genome, the authors should perform ChIP-seq of these histone marks to better understand the specific loci that are affected by Vitamin C deficiency.
2. The authors showed in Figure 6 that shRNA targeting KDM/JMJD family members affected Vitamin C-mediated epigenetic control of osteoblastic differentiation. What are the effects of overexpression of certain KDM/JMJD family members, which showed the best results in the shRNA experiment, on osteoblast differentiation? Promoting osteoblast differentiation like TET2 overexpression?
3. What are the relationship between Vitamin C-mediated 5hmC accumulation and H3K9me3/H3K27me3 downregulation? Do they cooperate or function independently to promote osteogenesis? This could be potentially answered by overexpression of TET2 in KDM/JMJD knockdown cells, or overexpression KDM/JMJD in TET2 knockdown cells.
4. Are the 5hmC-enriched regions in TET2 overexpression cells specific to bone-selective genes? Do they overlap with Vitamin C-induced 5hmC enrichment at bone-selective genes? The authors should perform hMeDIP-seq in TET2 overexpression cells to address this question.
5. How to explain the opposite effects of TET3 and TET2 on osteoblast differentiation? The authors

should at least discuss this in the discussion section.

6. To avoid off-target effects, the authors should use at least two shRNA or sgRNA sequences to confirm their results.

Minor points:

1. In Figure 1f, please move the labels “Bone60 gene-set” and “Fat76 gene-set” from the right side to the top of GSEA graphs.
2. In Figure 1e, the authors should label in the volcano plots with arrows showing the direction of up- or down-regulation with vitamin C deficiency, or at least label Log₂ fold change of (VitC+/- vs VitC+).
3. In Figure 1g, the authors didn't label the sample names in the heatmap.
4. Since gene sizes vary, the average 5hmC signal showed in Figure 2c is not giving the whole picture across the gene body. The authors should include TSS (transcription start site) and TES (transcription termination site) and normalize the data to the gene sizes.
5. Figure 2d and extended data figure 3d are very difficult to read, the authors should separate the genome browser tracks for VitC+ and VitC+/- like in Figure 2f (iii).
6. In the “Introduction” section, the background should be followed by a summary of the current study.
7. The 5hmC dot blot of VitC- NegCtrl shown in Figure 6d didn't show any signals, even the background signal for the highest loading concentration. The authors should double check on this.

RESPONSE TO REVIEWERS

Manuscript # NCOMMS-20-39677-T

Vitamin C epigenetically controls osteogenesis and bone mineralization

Thaler *et al.*

We sincerely thank the reviewers for expressing their interest in our study and for their time and efforts they spent in reviewing our manuscript. Their insightful and constructive comments, suggestions and thoughts were indeed very valuable to us and helped us to strengthen and refine our previously proposed epigenetic mechanisms through which Vitamin C orchestrates osteogenesis and bone health.

Over the last year, we comprehensively and diligently addressed the reviewers' comments and added extensive new *in vivo* and *in vitro* data which all support and strengthen our original work. In brief, we recapitulated key experiments in primary mouse and human bone marrow stromal cells which as suggested by **Reviewer 1**, increase the translational significance of our work. We also addressed **Reviewer 2's** concern centered around *in vivo* evidence of our proposed mechanism by characterizing Vitamin C-sufficient, bone-conditional *Tet1* and *Tet2* knockout mouse models which fully phenocopied the bone defect of our initial Vitamin C-deficient *Gulo* knockout mice. We also investigated Vitamin C's additional functions as antioxidant and modulator of alternative cell fates (including cellular senescence and retained cell stemness) during osteogenesis. Moreover, following **Reviewer 2's** and **3's** suggestions, we extended our gene knockdown and gene overexpression studies of Vitamin C-dependent histone and DNA demethylases during osteogenesis and conducted additional omics experiments including H3K9me3 ChIP-Seq and H3K27me3 ChIP-Seq as well as hMeDIP-Seq and RNA-Seq experiments.

These and additional experimentations resulted in 10 new figures, which expanded our manuscript to 10 main figures and 17 supplementary figures. We also comprehensively reworked our introduction and discussion sections to set our findings into context of the recent literature and to link them to clinical studies of patients with bone loss conditions and Vitamin C deficiency.

In summary, the newly presented, comprehensive experimental data reinforce our initial work and strengthen our conclusions that Vitamin C controls osteogenesis *in vitro* and *in vivo* by DNA hydroxylation and histone demethylation.

Below are point-by-point responses to the reviewers' comments. The reviewer points are in *italics* and our responses are in plain text.

Reviewer #1 (Remarks to the Author):

The manuscript "Vitamin C epigenetically controls osteogenesis and bone mineralization" is interesting, the aim is clear, and the methods seem appropriate. Previously similar studies have been published in different systems, but this study has some novel findings related to bone biology. The best part of this study is that the authors used an appropriate mice model (gulono-lactone oxidase,

GULO knockout) to study the role of vitamin C (as mice make vitamin C) in bone biology. The findings are novel in the field of bone biology.

The main drawback of this study is that authors performed *in vitro* studies on mice BMSCs, which lack translational importance. The author should perform one or two key experiments on human BMSCs (which are commercially available) to get some valuable translational data. The discussion of the manuscript needs to be elaborated. It does not tell the entire novel findings generated by authors. Overall, I am happy that reputed scientist Dr. Wijnen investigating vitamin C role in bone biology. There are few labs investigating the important role of vitamin C in musculoskeletal pathobiology.

Response: We would like to thank Reviewer 1 for the positive and supportive comments. We completely agree that the use of human BMSCs adds translational importance to our manuscript. In this extensively revised version of the manuscript, we comprehensively addressed this point and included experiments using human BMSCs from healthy donors for multiple experiments.

Briefly, we conducted osteogenic differentiation analyses on human BMSCs with and without Vitamin C, and performed similar experiments on human BMSCs during differentiation towards the adipogenic lineage. We find that reminiscent to murine cells, Vitamin C is critically required for osteogenic differentiation including osteogenic gene expression and is dispensable for differentiation of human BMSCs during adipogenic differentiation. Additionally, we included experiments in which we allowed Vitamin C-mediated osteogenic differentiation until day 35, followed by Vitamin C withdrawal until day 50. The results show that Vitamin C is constantly required throughout the osteogenic process, even at late stages of differentiation.

The new data are presented in new Supplementary Figures 4 and 5, and are described on page 9 (lines 1-12).

Additionally, by using the prolyl hydroxylase inhibitor 1,4-DPCA in human BMSCs we confirmed our conclusion that the collagenous matrix *per se* is not required for osteogenic gene expression. These results strengthen our notion that the epigenetic role of Vitamin C is critical for osteogenesis.

These new data are summarized in main Figure 5g,h and are described on page 10 (lines 1-5).

We also employed additional mouse models to increase *in vivo* relevance of our proposed epigenetic mechanisms involving TET-mediated 5hmC in bone. Briefly, we generated bone-selective conditional knockout mice of *Tet1*, *Tet2* or both. Bone-selective deletion was accomplished via *Prrx1-Cre*. The resultant experimental mice (*Tet1 flox/flox; Prrx1-Cre* and *Tet2 flox/flox; Prrx1-Cre* and *Tet1 flox/flox; Tet2 flox/flox; Prrx1-Cre*) were compared to *Prrx1-Cre* control mice. Importantly, all these mice were Vitamin C self-sufficient.

We found that deletion of *Tet1+Tet2* dramatically reduced global and local 5hmC levels and concomitant gene expression of bone-selective genes in femurs of these mice. μ CT on knockout bones demonstrate that *Tet1/2* mice suffer from severe bone deficiency as shown by strongly reduced bone volume, trabecular number and trabecular thickness as well as increased trabecular separation. Importantly, these aberrant skeletal parameters mirror the bone presentation of Vitamin C-deficient, TET-sufficient *Gulo*^{-/-} mice.

These additional data are summarized in new Figure 10 and are described on page 15 (lines 14-25) and page 16 (lines 1-10).

We extensively revised our manuscript and added a more detailed discussion and a more elaborated introduction which better discusses the clinical evidence for the role of Vitamin C and Vitamin C deficiency in skeletal health and disease. Importantly, in a clinical study involving post-menopausal women on estrogen replacement therapy, those women who received Vitamin C supplementation in addition to estrogen had increased bone mineral density compared to estrogen only recipients. These exciting observations suggest that Vitamin C supplementation could indeed complement the existing standard of care that is currently applied in a broad spectrum of degenerative bone conditions that benefit from bone anabolic strategies.

Reviewer #2 (Remarks to the Author):

The present manuscript entitled “Vitamin C epigenetically controls osteogenesis and bone mineralization” investigates the DNA and histone methylation basis of Vitamin C for bone integrity and osteoblastic differentiation. Although that Vitamin C regulation on osteogenesis and bone formation have been reported and that Vitamin C is known to act as an epigenetic regulator, this manuscript is the first to decipher detailed epigenetic mechanisms underlying effects of Vitamin C on osteogenesis. The major drawback is lack of enough in vivo experimental evidence to explore epigenetic targets.

Response: We appreciate Reviewer 2’s thoughtful, incisive, and critical feedback which encouraged us to study additional mouse models translating our *in vitro* findings into an *in vivo* context, to perform additional experiments using primary cells and to better articulate the approach and importance of our study. To address Reviewer 2’s constructive points, we generated extensive new data which resulted in 8 additional figures. We have improved the delivery of our introduction, results and conclusions which substantially advanced our study.

Following are the detailed comments:

Major concerns

1. The title proposes “epigenetically controls bone mineralization”, but lack of in vivo functional experiments on epigenetic targets. The authors should at least perform in vivo loss-of-function and gain-of-function analysis on the proposed target, Tet2.

Response: We thank the reviewer for raising this point as it clearly advanced our study. We agree that *in vivo* functional studies are highly relevant to ascertain our identified Vitamin C-mediated epigenetic mechanism essential for bone integrity.

To address this point, we generated bone-selective conditional knockout mice of *Tet1*, *Tet2* or both using a *Prrx1-Cre* driver. The resultant experimental mice (*Tet1 flox/flox; Prrx1-Cre* and *Tet2 flox/flox; Prrx1-Cre* and *Tet1 flox/flox; Tet2 flox/flox; Prrx1-Cre*) were compared to *Prrx1-Cre* control mice. Importantly, all these mice were Vitamin C self-sufficient.

We found that deletion of *Tet1+Tet2* dramatically reduced global 5hmC levels in bone tissue. Using hMeDIP-Seq, we performed a more detailed analysis on the 5hmC landscape with focus on bone-selective genes. By correlating hMeDIP-Seq data with matched RNA-Seq data, we found that almost all bone-selective genes (Bone60 gene set) lost 5hmC and reduced their gene expression in *Tet1/2* knockout femurs. Importantly, μ CT analysis demonstrate that the *Tet1/2* knockout mice suffer from bone deficiency as shown by strongly reduced bone volume, trabecular number and trabecular thickness as well as increased trabecular separation. Importantly, these aberrant skeletal parameters phenocopy the bone phenotype of Vitamin C-deficient but TET-sufficient *Gulo*^{-/-} mice.

These additional data are summarized in new Figure 10 and are described on page 15 (lines 14-25) and page 16 (lines 1-10).

Collectively, our revised manuscript provides strong *in vitro* and *in vivo* evidence that bone cell and bone tissue integrity is critically dependent on Vitamin C-dependent, TET-mediated activation of bone specific gene expression.

2. In this manuscript cell lines were used in several experiments instead of primary cells. It is risky to draw conclusions from cell lines particularly when the authors try to answer mechanistic questions related to in vivo phenomena. Primary bone marrow mesenchymal stem cells should be used throughout the manuscript for in vitro studies.

Response: We concur that the use of primary cells is more relevant for drawing conclusions and important for confirming data obtained from cell lines. Accordingly, we now present key experiments throughout our study using primary mouse or human BMSCs. In addition to our earlier mouse BMSC experiments assessing Vitamin C's role in osteogenic lineage maturation (now presented in Figure 4) or adipogenic differentiation (now Supplementary Figure 3) and investigations whether our epigenetic mechanisms are dependent on the ECM (now Figure 5) we added the following studies:

- 1) Using human BMSCs, we recapitulated the need for Vitamin C to drive osteogenic differentiation including osteogenic gene expression. Reminiscent to murine cells, Vitamin C is dispensable for differentiation of human BMSCs towards the adipogenic lineage Please, see new Supplementary Figures 4 and 5.
- 2) We conducted experiments evaluating the role of Vitamin C's antioxidant function in murine BMSCs (please see below, point #7, and addressed by new Supplementary Figure 6).
- 3) Similarly, the effects of Vitamin C on cellular senescence, proliferation as well as BMSC stemness markers, were performed in mouse BMSCs (please see below, point #6, addressed by new Supplementary Figure 7).
- 4) We repeated shRNA-mediated knock-down experiments for all 9 previously analyzed DNA and histone demethylases with 2 independent shRNA per gene (please see also below, point #4). Here we comprehensively analyzed the effects on the major osteogenic hallmarks: ECM deposition, alkaline phosphatase activity and ECM mineralization. These data are presented in new Supplementary Figure 13 and largely mimic the shRNA knockdown screening conducted earlier in cell lines. Again, we found that TET enzymes, particularly TET2, are essential for proper osteogenic differentiation and function.

For certain experiments such as the VP64-CRISPR/dCas9 mediated overexpression of the different DNA and histone demethylases, the use of primary cells was technically not feasible as this would require successful transfection of the VP64-CRISPR/dCas9 and gRNA vectors as well as drug-induced selection of the successfully transfected primary BMSCs before the initiation of experiments. Unfortunately, this would require multiple cell divisions which would compromise osteogenic potential and/or induce a senescent phenotype before the cells could be used for our actual experiments.

We also would like to point out that, although we completely agree that results gained from primary cells and tissues have higher relevance, our data obtained in MC3T3-E1 osteoblasts and MLO-A5 pre-osteocytes closely resemble data from primary cells and tissues. These similarities include:

- (i) General and specific epigenetic patterns around bone-specific genes and on different genomic regulatory elements including enhancers and super-enhancers
- (ii) Effects of Vitamin C- and demethylase-dependent gene expression patterns
- (iii) Relevance of Vitamin C and Vitamin C-dependent DNA and histone demethylases on major osteo-phenotypic markers like ECM deposition, ALPL activity and ECM mineralization (in the shRNA knockdown screening)

Therefore, by using tissues, cell lines and primary cells from mouse and human origin, our study provides strong and exhaustive evidence that Vitamin C controls osteoblastogenesis and bone integrity via epigenetic mechanisms.

3. Stem cell function should be examined using appropriate methods, such as multipotent differentiation and self-renewal assays.

Response: We thank the reviewer for bringing this point up as it allowed us to improve the explanation of our approach and experimental setting. We want to point out that there is strong divergence in terminology concerning BMSCs. A large body of literature in the past 2 decades discusses this issue, for example Ambrosi *et al.*, *Front Cell Dev Biol*, 2019, PMID 31572721; or Lindern *et al.*, *Transfus Med Hemother*, 2010, PMID 20737049; or Bianco *et al.*, *Cell Stem Cell*, 2008 PMID 18397751).

In our mouse study, we follow standard techniques that are commonly used in the bone field to isolate BMSCs. Briefly, we isolate bone marrow from femurs and tibias and propagate the adherent cell fraction to be used for osteogenic and adipogenic differentiation experiments. Therefore, our culture is a heterologous collection of stromal cell types. As we do not purify individual cell clones, but rather utilize the full, non-clonal culture, we decided to follow the current literature to define our culture as bone marrow stromal cells. We apologize for being unclear and erroneous in our initial manuscript but thank the reviewer again for bringing up this important point.

Importantly, these BMSC cultures are known to exhibit properties of multipotency towards, among others, the osteogenic and adipogenic lineages, however, this cell mixture is not considered a culture of true stem cells. Accordingly, these cultures are known to have limited self-renewal capacity.

Similarly, we purchased human BMSCs, which were isolated from human bone marrow and cultured for 2 passages before sale. Although the manufacturer describes these cells as human mesenchymal stem cells, we believe it is more appropriate to call them bone marrow stromal cells according to the above criteria.

That being said, in our manuscript we have included experiments on multipotent differentiation on our BMSC cultures. Mouse BMSC differentiation towards the osteogenic lineage and adipogenic lineage are presented in Figure 4 and Supplementary Figure 3, respectively (described on page 7, lines 22-25 and page 8 lines 1-23). Differentiation of human BMSCs along the osteogenic and adipogenic lineage are displayed in the new Supplementary Figures 4 and 5 (described on page 9, lines 1-12). We note that these experiments do not truly evaluate multipotency of clonal stem cells, but rather these studies were designed to evaluate how Vitamin C modulates osteogenic or adipogenic differentiation.

4. It is necessary to verify epigenetic targets in primary mesenchymal stem cells.

Response: We agree with the reviewer that use of primary cells increases the significance of our results. We now include an important set of new experiments where we knocked down all 9 analyzed DNA and histone demethylases in primary mouse BMSCs using 2 independent shRNAs per gene. We comprehensively analyzed the major osteogenic hallmarks: ECM deposition, alkaline phosphatase activity and ECM mineralization.

Reminiscent to our knockdown screening in MC3T3-E1 and MLO-A5 cell lines, we found that TET enzymes, TET1 and particularly TET2, are essential for proper osteogenic differentiation and function as measured by impaired ECM deposition, mineralization and ALPL activity after *Tet2* depletion. Also, we found that knockdown of H3K9me3 demethylases *Kdm4a* and *Kdm4b* resemble *Tet2* loss except for in ALPL activity, while *Kdm6a* and *Kdm6b* are mainly important for late stages of differentiation, for ECM mineralization.

Taken together, the results in primary cells confirm and reinforce that TET enzymes, particularly TET2, are essential for proper osteogenic differentiation and function.

These data are presented in the new Supplementary Figure 13 and described on page 14 (lines 7-12)

5. The authors should strengthen the significance of Vitamin C epigenetically promoting bone formation regarding clinical relevance. For example, in the case of bone defects, can Vitamin C regulate epigenetic mechanisms to improve bone healing? Or can Vitamin C be used to epigenetically promote the stem cell function and increase the therapeutic effects?

Response: These are all interesting questions! Our primary objective of the current study was to establish the epigenetic basis of Vitamin C's role on osteoblastogenesis and bone formation. For many decades, the requirement for collagen fiber assembly has been considered the main role of Vitamin C in bone tissue, yet its recently emerging epigenetic actions have not been investigated in the bone field. Our study provides clear evidence that Vitamin C's epigenetic role is critical throughout the osteogenic

lineage, while the collagen matrix deposition in bone via collagen hydroxylation appears to be mechanistically downstream from epigenetic control.

Our Vitamin C-insufficient *Gulo* knockout mice strikingly recapitulate clinical bone parameters of human Vitamin C deficiency and scurvy. We observed that these defects correlate with impaired epigenetic features. Consistent with this, bone-specific deletion of TET enzymes impairs not only the 5hmC landscape but also causes skeletal defects despite the availability of Vitamin C in these mice. Importantly, bone phenotypes from *Tet1/2* knockout mice mirror *Gulo* knockout bones and, by extension therefore, human bone symptoms during Vitamin C deficiency.

The reviewer's questions are clearly important and interesting, but (pre-)clinical studies would be outside of the scope of the current manuscript. However, we have extended our analyses to primary human BMSCs which fully recapitulated our earlier results in murine cells (please see also point #1 and #4). Follow-up studies on the clinical relevance of Vitamin C, for example during bone loss conditions, are conceivable and will be important to translate our findings into clinical practice.

That being said, current literature underlines the importance of adequate Vitamin C levels to human bone health and bone fracture risk. Please, see our new introduction for a comprehensive overview. Importantly, in a clinical study involving post-menopausal women on estrogen replacement therapy, those women who received Vitamin C supplementation in addition to estrogen had increased bone mineral density compared to estrogen only recipients. These exciting observations suggest that Vitamin C supplementation could indeed complement the existing standard of care that is currently applied in a broad spectrum of degenerative bone conditions that benefit from bone anabolic strategies.

6. The results show that Vitamin C promotes osteogenic differentiation. However, it might actually be attributed to a general effect of Vitamin C recovering stemness of cells, as previously reported the pluripotency reprogramming function of Vitamin C. The authors need to check if broader functions of stem cells are affected, such as proliferation, senescence, cell cycle progression and expression of stemness genes.

Response: The reviewer's point is very valid. We would like to refer again to point #3 concerning the identity of our BMSC cultures as a non-clonal, heterologous culture of stromal cells. Nonetheless, the reviewer's point that Vitamin C may modulate alternative cell fates such as cellular senescence or affects the cell's proliferative capacity which collectively can affect osteoblastogenesis is legitimate.

Therefore, we assessed the induction of cell cycle arrest and cellular senescence in primary mouse BMSCs. We evaluated the activity of senescence-associated β -galactosidase, proliferation and expression of cell cycle arrest and senescence markers, *p16*, *p19* or *p21*. We did not find evidence that Vitamin C alters cell proliferation or induction of cellular senescence. These new results are displayed in the new Supplementary Figure 7a-c and described on page 10 (lines 15-21).

Consistent with the notion that our BMSC cultures are stromal cells, we found that, unlike mouse embryonic stem cells, classical stem cell markers such as *Oct4*, *Sox2* or *Klf4* are hardly expressed in

murine BMSCs. Additionally, temporal gene expression patterns of BMSC markers were largely unaffected during Vitamin C-mediated osteogenic differentiation. These new data are presented in Supplementary Figure 7d-f and described on page 10 (lines 21-25) and page 11 (lines 1-5).

We also note that Vitamin C's pro-osteogenic properties are also still apparent in differentiated osteogenic cells (see our earlier data in Figure 4f-g and our new data in Supplementary Figure 4b,c). In both instances Vitamin C withdrawal during osteogenic differentiation of mouse and human BMSCs at the osteoblastic/early osteocyte stage (day 35) completely abrogates further differentiation as measured by for example *Dmp1* activity, ECM mineralization and osteogenic gene expression. Only Continuous Vitamin C supplementation guarantees further differentiation through day 50 when main osteocyte markers like *Sost* are expressed. These data suggest that Vitamin C is stringently required for all stages of osteogenic differentiation. This data is discussed on page 8 (lines 10-14) and page 9 (lines 1-6). Collectively, we conclude that the pro-osteogenic properties of Vitamin C are unlikely due to modulated cellular senescence, proliferation or retained stemness.

7. The authors have proved that effects on collagen formation are dispensable for Vitamin C promoting osteogenesis. However, Vitamin C has also been recognized as a potent antioxidant. Whether the antioxidant effects actually underlie the effects of Vitamin C? Controls such as NAC should be used in vivo experiments and on cultured stem cells. It would be better if the authors perform supplemental experiments on loss-of-function experiments of antioxidant mechanisms.

Response: Indeed, a main and very well studied function of Vitamin C is its role as scavenger against free radicals. To dissect the epigenetic role from the antioxidant functions of Vitamin C, we now performed experiments in which we compared the osteogenic potency of Vitamin C with another antioxidant, NAC (N-Acetyl-L-Cysteine) in primary mouse BMSCs. The data clearly show that only Vitamin C is able to induce osteogenic differentiation as well as DNA hydroxymethylation, H3K9me3 demethylation, and H3K27me3 demethylation. These observations suggest that the antioxidant function of Vitamin C is independent from its pro-osteogenic role. These data are presented in new Supplementary Fig. 6 and are described on page 10, line 7-15.

Importantly, in complementary experiments, we found that inhibiting the catalytic reaction that Vitamin C-dependent epigenetic modulators depend on (α -KG to Succinyl-CoA) clearly abolishes the epigenetic functions of VitC in differentiating primary mouse BMSCs as well as in differentiating MC3T3-E1 osteoblasts (please, see new Figure 6). These results indicate that the major function of Vitamin C during osteogenic differentiation is related to its activity as a co-factor in the TCA cycle and therefore as co-factor for epigenetic modulators, including TET enzymes and histone demethylases.

8. It is interesting that Vitamin C is dispensable for adipogenesis, as osteogenesis and adipogenesis is often reversely correlated. More discussions are needed to explain the sole requirement of Vitamin C for osteogenesis and why the epigenetic changes do not affect adipogenesis.

Response: We fully agree with the reviewer that this is an interesting finding, especially considering that both lineages can be derived from the same BMSC culture. As requested, we strengthened the discussion to include tissue type-specific needs of Vitamin C, including the osteogenic versus adipogenic lineage. Please, see page 18 (line 7-12).

9. Although histone methylation is tightly regulated by Vitamin C according to this study, the mechanisms remain unclear. Only Tet2 is revealed responsible for the DNA methylation changes. It is recommended to further dissect mechanisms of Vitamin C regulating histone methylation together with functional experiments of the targets.

Response: Thank you for raising this point. In our studies, TET enzymes and 5hmC generation has stood out as particularly important for osteogenesis. However, we find that Vitamin C-dependent histone demethylases contribute to osteoblastogenesis, and we agree with the reviewer's point that their role during osteogenic differentiation benefited from a better assessment and discussion.

We have now conducted additional experiments on H3K27me3 and H3K9me3 histone marks as well as their Vitamin C-dependent demethylases of the Jumonji-C domain-containing histone demethylase family (JHDMs/KDMs). Further, we performed H3K9me3 and H3K27me3 ChIP-Seq in the presence or absence of Vitamin C during osteogenesis.

We thoroughly analyzed the profiles of these markers around osteogenic loci and at distal intergenic sites and correlated the results with RNA-Seq data as well as with GREAT-GO phenotypical annotations. Briefly, we found that Vitamin C is needed to demethylate H3K27me3 residues around transcription start sites and gene bodies of bone-specific genes. However, unlike for 5hmC, overall bone-specific gene expression did not correlate with altered H3K9me3 or H3K27me3 levels around these genes and their promoters. Only a subset of bone-specific genes was expressed in a Vitamin C-dependent, H3K9me3-dependent fashion, such as *Phospho1* and *Dmp1*, or in a Vitamin C-dependent, H3K27me3-dependent fashion including *Siglec15*, *Bmp8b* and *Notch3*. These data are presented in the new Supplementary Figures 10 and 11 and described on page 12 (lines 6-24).

In addition, we not only knocked down Vitamin C-dependent H3K9me3 and H3K27me3 demethylases (please see point #2 and #4 above and Figure 7, Supplementary Figure 12 and new Supplementary Figure 13) but also stably overexpressed these genes via VP64-CRISPR/dCas9 technology using 2 independent gRNAs. We also included combinations in which we, for example, knocked down or overexpressed members of the same family during osteogenic differentiation. Functionally, we then assessed osteogenic hallmarks ECM deposition, ALPL activity, ECM mineralization as well as mRNA expression of several osteogenic genes. We find that overexpression of H3K9me3 or H3K27me3 demethylases had only minor consequences for osteoblastogenesis, with only KDM4C or KDM6A causing some enhancement in osteoblastic differentiation. This was in stark contrast to TET2 overexpression that clearly facilitated osteoblastogenesis. We wondered whether the pro-osteogenic effect of TET2 could be potentiated by co-expressing KDM4C or KDM6A and vice versa. However, we found that none of the combinatorial conditions surpasses the pro-osteogenic properties of individual genes. Gene overexpression data are presented in the new Supplementary Figures 14, 15 and 16 and described on page 14 (lines 12-24).

It is conceivable that while TET-mediated 5hmC generation acts as an epigenetically active mark designated to facilitate gene expression, demethylation of H3K27me3 or H3K9me3 merely reduces or neutralizes repressive epigenetic marks; and without establishing activating signals such as H3K27ac or H3K9ac, gene expression may still be restricted. Indeed, histone acetyltransferases which would generate such activating epigenetic marks have not been described as Vitamin C-dependent. The latter rationalization may explain why we find 5hmC and 5hmC-generating enzymes to orchestrate a variety of physiological aspects during osteogenic differentiation, while the relevance of histone demethylases is more restricted.

10. The introduction and discussion sections lack comprehensive review of previous studies, particularly the application of Vitamin C in stem cell and bone contexts, its role as an epigenetic regulator and the understanding of Tets in regulating mesenchymal stem cell function and bone homeostasis. The study should be based on clear and sound references of the current advance.

Response: The manuscript was comprehensively rewritten and in this process the references were expanded and updated accordingly. We have advanced our introduction to briefly summarize the current literature describing clinical aspects of Vitamin C for bone health and homeostasis. We have also expanded the introduction and discussion to include the current literature on 5hmC, TETs, their tissue selectivity and importance during differentiation of many lineages. Please, see page 4 (lines 24-25), page 5 (lines 1-5), page 17 line 24 to page 18 line 12. In the relevant result section, we also included information and references on Vitamin C's function as antioxidant and during reprogramming of induced pluripotent stem cells. Please, see page 10 (lines 7-25) and page 11 (lines 1-5).

Minor concerns

1. There is no functional experiment data about 5hmC, so page5/line104 "by the activating 5hmC" should be changed to "with activating 5hmC".

Response: We thank the reviewer for their careful assessment of the manuscript. As the manuscript was extensively rewritten, this particular phrase was removed.

2. More than half of the references are published at over five years ago and the knowledge of the latest advances in the field is lacking.

Response: During the revision process and reformatting the manuscript, also the references were expanded and updated, and now reflect the current state of the field.

Reviewer #3 (Remarks to the Author):

NCOMMS-20-39677-T, Thaler and co-authors, Vitamin C epigenetically controls osteogenesis and bone mineralization

In this study, by using vitamin C-deficient Gulo knockout mouse model, the authors showed that Vitamin C deficiency induces transcriptional changes in bone tissues, with vitamin C-deficient mice displaying a phenotype of severe defects in bone structure. Vitamin C controls the epigenetic program in bone tissue and during osteogenic differentiation by downregulating the levels of repressive histone marks H3K9me3 and H3K27me3, and promoting Tet-mediated 5hmC deposition at bone-specific genes. Vitamin C is required for bone marrow-derived mesenchymal stem cells (BMSCs) differentiation into osteoblast lineage, but not adipocyte lineage. Further, the authors showed that knockdown of TET2 and certain members of KDM/JMJD family demethylases affects Vitamin C-mediated osteogenic differentiation, while overexpression of TET2 promotes osteoblast differentiation.

Overall, the authors presented an interesting study about the epigenetic function of Vitamin C during osteogenic differentiation. The following are some comments:

Response: We were pleased to hear that the reviewer finds our study interesting. We would like to thank the reviewer for their positive and constructive feedback and comments which clearly helped to improve our manuscript and strengthened our conclusions.

Major points:

1. The authors showed by western blot that vitamin C deficiency promotes the accumulation of the repressive histone marks H3K9me3 and H3K27me3, however, this only provides data of the overall levels of these histone marks in the genome, the authors should perform ChIP-seq of these histone marks to better understand the specific loci that are affected by Vitamin C deficiency.

Response: Thank you for raising this important point. We conducted H3K9me3 and H3K27me3 ChIP-Seq data in differentiating osteoblasts with and without Vitamin C (new Supplementary Figures 10 and 11). We evaluated these two repressive histone marks around the TSS and over gene bodies and also integrated ChIP-Seq data with corresponding RNA-Seq data.

We found that Vitamin C supplementation causes H3K27me3 demethylation around transcription start sites and gene bodies of bone-specific genes, however, in contrast to 5hmC, overall bone-specific gene expression did not correlate with altered H3K9me3 or H3K27me3 levels at these genes. Only a subset of bone-specific genes is expressed in a Vitamin C-dependent, H3K9me3-dependent fashion, such as *Phospho1* and *Dmp1*, or in a Vitamin C-dependent, H3K27me3-dependent fashion including *Siglec15*, *Bmp8b* and *Notch3*. Closer examination of the genomic distribution of these epigenetic marks shows that a large proportion (37-70%) of H3K9me3, H3K27me3 or also 5hmC peaks is located in distal intergenic regions. GREAT analyses on these distal loci where Vitamin C most significantly altered these marks, revealed that only distal 5hmC-marked sites strongly correlate with bone-associated phenotypes. H3K27me3-occupied loci correlated with neurological phenotypes while there was no significant enrichment of phenotypes for distal H3K9me3 related loci. Thus, we conclude that Vitamin C-mediated

H3K27me3 and H3K9me3 demethylation is relevant for the expression of a subset of osteoblastic genes, while 5hmC broadly alters bone-relevant loci to promote their gene expression.

These data are presented in the new Supplementary Figures 10 and 11 and are described on page 12 (line 6-24)

2. The authors showed in Figure 6 that shRNA targeting KDM/JMJD family members affected Vitamin C-mediated epigenetic control of osteoblastic differentiation. What are the effects of overexpression of certain KDM/JMJD family members, which showed the best results in the shRNA experiment, on osteoblast differentiation? Promoting osteoblast differentiation like TET2 overexpression?

Response: We agree with the reviewer that this is an interesting point, Reviewer 2 had a similar point (see Reviewer 2, point #9).

We expanded our VP64-CRISPR/dCas9-mediated overexpression studies to H3K9me3 demethylases (KDM4A, KDM4B and KDM4C) and H3K27me3 demethylases (KDM6A, KDM6B, JHDM1D). Each overexpression was performed with 2 independent gRNAs and we assessed for bone hallmarks ECM deposition, ECM mineralization, ALPL activity and pro-osteogenic gene expression as functional readouts.

We found that overexpression of H3K9me3 or H3K27me3 demethylases had only minor consequences for osteoblastogenesis, with only KDM4C or KDM6A causing some enhancement in osteoblastic differentiation. This was in stark contrast to TET2 overexpression that facilitated osteoblastogenesis. We wondered whether the pro-osteogenic effect of TET2 could be potentiated by co-expressing KDM4C or KDM6A and vice versa (please, also see the next point #3). However, we found that none of the combinatorial conditions surpasses the pro-osteogenic properties of individual genes. Gene overexpression data are presented in the new Supplementary Figures 14, 15 and 16 and described on page 14 (lines 15-24).

It is conceivable that while TET-mediated 5hmC generation acts as an epigenetically active mark designated to facilitate gene expression, demethylation of H3K27me3 or H3K9me3 merely reduces or neutralizes repressive epigenetic marks; and without establishing activating signals such as H3K27ac or H3K9ac, gene expression may still be restricted. Indeed, histone acetyltransferases which would generate such activating epigenetic marks have not been described as Vitamin C-dependent. The latter rationalization may explain why we find 5hmC and 5hmC-generating enzymes to orchestrate a variety of physiological aspects during osteogenic differentiation, while the relevance of histone demethylases is more restricted.

3. What are the relationship between Vitamin C-mediated 5hmC accumulation and H3K9me3/H3K27me3 downregulation? Do they cooperate or function independently to promote osteogenesis? This could be potentially answered by overexpression of TET2 in KDM/JMJD knockdown cells, or overexpression KDM/JMJD in TET2 knockdown cells.

Response: This is indeed an interesting question. To address this point, in addition to our knockdown screening, we first overexpressed all 9 candidate epigenetic modulators individually and evaluated their impact on osteogenic differentiation (please, see point #2). Conditions that showed a significant impact on any parameter during osteogenic differentiation consistently with both gRNAs or shRNAs were subjected to combinatorial gene knockdown or gene overexpression studies.

For gene knockdown studies, we co-depleted enzymes of the same family pairwise (for example, *Kdm4a+Kdm4b* knockdown) or all three enzymes simultaneously (for example, *Kdm4a+Kdm4b+Kdm4c* knockdown). We found that co-depletion of at least two H3K27me demethylase family members (e.g. *Kdm6a* and *Kdm6b*) potentiates the deleterious effect on some parameters during osteoblastic differentiation in MC3T3-E1 or MLO-A5 cells suggesting that these enzymes are redundant and may compensate to some extent during osteogenesis. We note that we performed shRNA depletion experiments on individual genes and in combination in three different cell types during osteogenic differentiation (MC3T3-E1 pre-osteoblast, MLO-A5 pre-osteocytes and primary mouse BMSCs) with similar results. Results of knockdown studies are presented in Figure 7, and Supplementary Figures 12 and 13 and described on page 13 (lines 1-25) and page 14 (line 1-5).

For overexpression studies, we only found TET2, KDM4C and KDM6A to promote osteogenic differentiation parameters. We then evaluated their pairwise overexpression or overexpressed all 3 genes simultaneously. However, we found that none of the combinatorial conditions surpasses the pro-osteogenic properties of individual genes. However, on the contrary, we observed that overexpression of all 3 genes (or when TET2 and either histone demethylase where co-overexpression) performed worse than individual gene overexpression for some osteogenic parameters. For example, KDM6A overexpression robustly enhances ALPL activity while TET2 or KDM4C do not. The combinatorial overexpression of all 3 genes does not show increased ALPL activity, suggesting that the KDM6A-induced ALPL activity is neutralized by TET2 or KDM4C co-expression. While much additional mechanistic investigations are needed to dissect the individual contribution of each epigenetic modulator on each osteogenic hallmark, we collectively suggest that some redundancy exists within the same enzyme family and that TET2 is both necessary for osteogenic differentiation and sufficient to enhance osteoblastogenesis. Gene overexpression data are presented in the new Supplementary Figures 14, 15 and 16 and described on page 14 (lines 15-24).

4. Are the 5hmC-enriched regions in TET2 overexpression cells specific to bone-selective genes? Do they overlap with Vitamin C-induced 5hmC enrichment at bone-selective genes? The authors should perform hMeDIP-seq in TET2 overexpression cells to address this question.

Response: To further assess the relevance of TET2 on 5hmC in osteogenic cells, we generated mouse models with bone conditional deletion of *Tet1*, *Tet2* or both using *Prrx1-Cre*; all these mice were Vitamin C self-sufficient. *Tet1+Tet2* double knockout mice had severe bone defects that mimicked Vitamin C-deficient *Gulo* knockout mice. We found that deletion of *Tet1+Tet2* also dramatically reduced global 5hmC levels in bone. Using hMeDIP-Seq, we performed a more detailed analysis on 5hmC landscape in bone tissue. We also correlated hMeDIP-Seq with matching RNA-Seq analyses to evaluate the resultant

gene expression changes. We found that almost all bone-selective genes (Bone60 gene set) including *Sp7* and *Dlx3* lost 5hmC and reduced their gene expression in *Tet1/2* knockout bone. These 5hmC and gene expression adaptations also mirror our earlier hMeDIP-Seq and RNA-Seq data from Vitamin C-deficient *Gulo* knockout mice.

These data are displayed in the new Figure 10 and described on page 15 line 14 to page 16 line 10.

5. *How to explain the opposite effects of TET3 and TET2 on osteoblast differentiation? The authors should at least discuss this in the discussion section.*

Response: We agree with the reviewer that this is indeed an interesting observation. We included a discussion of these opposing effects of TET2 and TET3 in the discussion section. We hypothesize that the role of TET3 diverges from TET1 and TET2 during osteogenesis and that each enzyme fulfills a distinct role. It is possible that the specificity of TET3 during osteogenesis differs from other TET enzymes, for example by controlling 5mC hydroxylation of alternative, non-bone-specific sites and targets gene and thus controlling different cellular functions. Please see the discussion on page 17 (line 17-22).

6. *To avoid off-target effects, the authors should use at least two shRNA or sgRNA sequences to confirm their results.*

Response: We understand and agree with the reviewer on this remark. In the first set of experiments that were presented in the initial submission, we performed an efficiency screen for each gRNA or shRNA beforehand and then pooled the 2 most efficient gRNAs/shRNAs for cell transfections. In all new experiments, the two best gRNAs/shRNAs were transfected separately and all new results are presented showing the effects of each single gRNAs/shRNAs (new Supplementary Figures 13 to 16). These data show similar results as in the first set of data with the pooled gRNAs/shRNAs. We further clarify our approach in the methods section and in the figure legends.

Minor points:

1. *In Figure 1f, please move the labels "Bone60 gene-set" and "Fat76 gene-set" from the right side to the top of GSEA graphs.*

Response: We appreciate this comment and have moved the labels as suggested. We note that GSEA analysis of the Fat76 gene set is now located in Supplementary Figure 2b.

2. *In Figure 1e, the authors should label in the volcano plots with arrows showing the direction of up- or down-regulation with vitamin C deficiency, or at least label Log₂ fold change of (VitC^{+/-} vs VitC⁺).*

Response: We agree with the reviewer. We have now improved the annotation of the x-axis of the volcano plots in Figure 1e and Supplementary Figure 2a.

3. In Figure 1g, the authors didn't label the sample names in the heatmap.

Response: We thank the reviewer for pointing out this mishap. We have now correctly labeled the heatmap in Fig. 1g.

4. Since gene sizes vary, the average 5hmC signal showed in Figure 2c is not giving the whole picture across the gene body. The authors should include TSS (transcription start site) and TES (transcription termination site) and normalize the data to the gene sizes.

Response: We understand and agree with the reviewers point. This type of analysis was performed with the CEAS software. This software normalizes all genes to a hypothetical length of 4 kb where the 4 kb mark represents the TES for each gene. Thus, to avoid misunderstandings, we now changed the labeling of the x-axis of former Figure 2c (now new Figure 3a) and replaced the former annotations with TSS and TES. The same change was performed in subsequent figures throughout the manuscript.

5. Figure 2d and extended data figure 3d are very difficult to read, the authors should separate the genome browser tracks for VitC+ and VitC+/- like in Figure 2f (iii).

Response: In the former Figure 2d (and now new Figure 3b) and similar figures, we overlaid the tracks for the different treatments so that the differences between groups more clearly visible. We tried to separate the tracks, but it became increasingly difficult to visually compare matching 5hmC loci. If further follow up on this point is needed, we are happy to provide separated tracks as an additional supplementary figure if space allows.

6. In the "Introduction" section, the background should be followed by a summary of the current study.

Response: We appreciate this comment. The manuscript was comprehensively rewritten, and we now included a study summary at the end of the introduction.

7. The 5hmC dot blot of VitC- NegCtrl shown in Figure 6d didn't show any signals, even the background signal for the highest loading concentration. The authors should double check on this.

Response: We thank the reviewer for their careful assessment of the manuscript. NegCtrl samples at day 0 have a very low 5hmC signal. This is particularly clear when comparing day 0 samples to sample that were treated with Vitamin C and subsequently showed massively increased 5hmC levels over 28 days of differentiation. The blot impressively shows the extent of Vitamin C-mediated 5hmC generation during osteoblastic differentiation. Nonetheless, we have double checked the dot blot in Figure 6d (now Figure 7e) and verified its authenticity. We increased the contrast of the blot in the new figure, so now the minimal 5hmC levels at day 0 become slightly visible in the lowest dilution.

Reviewers' comments:

Reviewer #1 (Remarks to the Author):

The author addressed my as well as other reviewers' comments successfully. They added new in vivo and in vitro data which strengthened the work. The author used primary mouse and human bone marrow stromal cells to perform additional work. They also performed in vivo studies using bone-conditional Tet1, and Tet2 knockout mouse models.

Reviewer #2 (Remarks to the Author):

I am satisfied with the authors' revision. Still some minor comments are as follows:

1. Representative pictures of control and knockout mice (e.g., body size), should be added in Figure 10 and Supplementary Figure 1. Previous studies indicated that there was a difference in the body size between the Vitamin C-deficient Gulo knockout mice and control mice. Thus, the comparisons of body size data are beneficial to the understanding of readers.
2. The pictures (VitC- group) of Alizarin Red staining are less plausible, because of the whiteness.

Reviewer #3 (Remarks to the Author):

The authors have performed lots of new experiments and addressed my comments/concerns. With the responses to the comments/concerns from the other reviewers, I think the manuscript has been significantly improved.

RESPONSE TO REVIEWERS

Manuscript # NCOMMS-20-39677-T

Vitamin C epigenetically controls osteogenesis and bone mineralization

Thaler *et al.*

We would like to take the opportunity to thank the reviewers for evaluating the revised version of our manuscript and are delighted that they were satisfied with our new results. We have addressed the few remaining minor comments as detailed below.

REVIEWERS' COMMENTS

Reviewer #2 (Remarks to the Author):

We thank reviewer #2 for its careful review of our work and address its comments below.

I am satisfied with the authors' revision. Still some minor comments are as follows:

1. Representative pictures of control and knockout mice (e.g., body size), should be added in Figure 10 and Supplementary Figure 1. Previous studies indicated that there was a difference in the body size between the Vitamin C-deficient Gulo knockout mice and control mice. Thus, the comparisons of body size data are beneficial to the understanding of readers.

We now include representative whole body radiographic images in supplementary figure 1. We note that the Gulo knockout mice were already 15 weeks old when we started vitamin C deprivation, thus whole-body size was not affected by vitamin C withdrawal. However, as the comparison clearly shows, major manifestations like scoliosis are clearly visible after 5 weeks of vitamin C absence. For the conditional Tet1/2 knockout mice, differences in body size between these mice and control mice were not really observable, thus we did not include pictures for these mice in figure 10.

2. The pictures (VitC- group) of Alizarin Red staining are less plausible, because of the whiteness.

In regard to the whiteness observed in the VitC- groups for the Alizarin red images, this is due to the total lack of matrix mineralization when vitamin C is absent. The scanning of the plates made the whiteness even more dominant. However, if zoomed in, the unmineralized cell layer becomes visible in some of the vitamin C negative samples where the whiteness is not too pronounced.